# Triethylamine vapor-induced cyclization reaction in cocrystals leading to cocrystal-to-polycrystal transformation

Ling Zhu[1], Xiaoli Zhao[1], Yanfei Niu[1], Lianrui Hu [1] ✉, Weitao Dou [1], Hai-Bo Yang [1,2], Lin Xu [1,2] ✉ & Ben Zhong Tang [3] ✉

The structural transformation of crystals triggered by external stimuli is a fascinating area in materials science and supramolecular chemistry. Despite the potential of organic reactions driving crystal conversions, its exploration remains limited, primarily due to challenges in maintaining crystallinity during significant structural changes. In this study, we present an intriguing example of a triethylamine vapor-induced cyclization reaction in a cocrystal, leading to its conversion into polycrystals. Initially, a charge transfer cocrystal with hydrogen-bond interactions denoted as AOTC, was prepared from 9-anthracene-substituted indolino-oxazolidine (Box) in its open form and 1,2,4,5-tetracyanobenzene (TCNB). Treatment of cocrystal AOTC with triethylamine vapor effectively induced a cyclization reaction, resulting in the formation of single-crystal AIC (closed form Box) and cocrystal ACTC (a cocrystal of AIC and TCNB), both of which were suitable for direct X-ray single crystal diffraction analysis. Experimental and theoretical analysis revealed that the cocrystal-to-polycrystal transformation was primarily driven by the cyclization reaction and the synergistic effects of intermolecular D−A and C−H···N hydrogen bond interactions. Additionally, this unique base-induced transformation was utilized for information storage applications. This research not only provides a rare instance of cocrystal-to-polycrystal transformation through a simple yet effective approach but also offers a strategy for crystal engineering.

Crystal-to-crystal transformations with solid-state structural changes are fascinating phenomena that offer opportunities to form unusual products[1–3] and visualize reaction processes[4]. In the past few decades, crystal phase transitions have been reported under exposure to various external stimuli such as heat[5–7], light[8–10], pressure[11–13], and solvent vapors[14–16]. These transitions can induce color changes[17], anisotropic shrinking/elongation of crystal size[18], fluorescence emission switching[19], etc[20,21]. Moreover, materials undergoing crystal transitions have demonstrated significant potential across diverse fields, including sensors, soft robotics, energy harvesting, and information storage[22–25]. However, compared to liquid or gas phases, solid-state crystal transformations are more constrained due to the restricted molecular movement within the crystal lattice, which limits the number of organic reactions that can occur[26,27]. Additionally, crystals tend to fragment easily during significant structural changes caused by reactions[28–30]. Therefore, driving crystal transitions through organic reactions while retaining crystallinity, especially when transforming into polycrystals, is particularly challenging.

[1]State Key Laboratory of Petroleum Molecular & Process Engineering, Shanghai Key Laboratory of Green Chemistry and Chemical Processes, School of Chemistry and Molecular Engineering, East China Normal University, Shanghai, China. [2]Hainan Institute of East China Normal University, Sanya, China. [3]School of Science and Engineering, Shenzhen Institute of Molecular Aggregate Science and Technology, The Chinese University of Hong Kong, Shenzhen (CUHK-Shenzhen), Shenzhen, Guangdong, China. ✉e-mail: lrhu@chem.ecnu.edu.cn; lxu@chem.ecnu.edu.cn; tangbenz@cuhk.edu.cn

Recently, organic cocrystals have garnered considerable attention due to their unexpected and versatile chemicophysical properties[31], which have enabled their use in various applications, such as photo-thermal conversion[32,33], near-infrared emission[34], semiconductor[35] and optical waveguide[29,36,37], among others[38–41]. Cocrystal engineering provides a platform for exploring the abundant transformation processes of multi-component crystals under various external stimuli[42–46]. For example, Liu and colleagues reported perylene-TCNB cocrystals exhibiting abnormal piezochromic luminescent behaviors[11]. Hu's group developed cocrystals capable of macroscopic mechanical bending upon repeated stimulation with THF solvent[15] (Fig. 1a). However, these transformation processes only involve absorption or desorption of solvents, leading to changes in donor-acceptor (D–A) arrangements within the cocrystal. In 2023, Stoddart and co-workers reported cocrystal-to-cocrystal transformation with donor exchange, albeit only successfully achieved at the microcrystalline scale (film)[19] (Fig. 1a). Therefore, challenges persist regarding cocrystal transformations, especially those driven by organic reactions. Particularly considering that the different non-covalent intermolecular interactions may present in the product crystals compared to those in the reactants, which may lead to the formation of a cocrystal with a different stoichiometric ratio. Theoretically, under external stimuli, a cocrystal with a specific stoichiometric ratio could transform into multiple cocrystals with different stoichiometric ratios, known as cocrystal-to-polycrystal transformation. Among a diverse range of crystal transformations, cocrystal-to-cocrystal transformations remain at a nascent stage of development compared to single-crystal-to-single-crystal transformation[47], and cocrystal-to-polycrystal transformations are even rarer.

The indolino[2,1-b]oxazolidine (Box) derivatives molecular switches possess two metastable states, commonly referred to as the open and closed form, with respect to the status of the oxazolidine ring[48,49]. Base treatment of Box can induce the conversion from the open to the closed form (Fig. 1b)[50,51]. Here, we designed an open-form 9-anthracene-substituted indoline-oxazolidine molecule (AIO), which incorporates indolino as a reactive group, chloride ions as hydrogen bond acceptors and anthracene as an electron-donor moiety (Fig. 1c). Subsequently, a charge-transfer cocrystal with hydrogen bonds, named AOTC, was obtained by the cocrystallization of AIO with 1,2,4,5-tetracyanobenzene (TCNB) in a 2:1 ratio. We envisaged that triethylamine (NEt₃), a commonly used organic liquid base with a low boiling point of 89 °C[52], could be utilized as a solvent vapor to induce oxazolidine ring closure in the AOTC cocrystal. The change in molecular structure intriguingly led to the transformation of the AOTC cocrystal into single crystals containing only the closed-form Box (AIC), as well as cocrystals of AIC and TCNB (named ACTC), demonstrating a

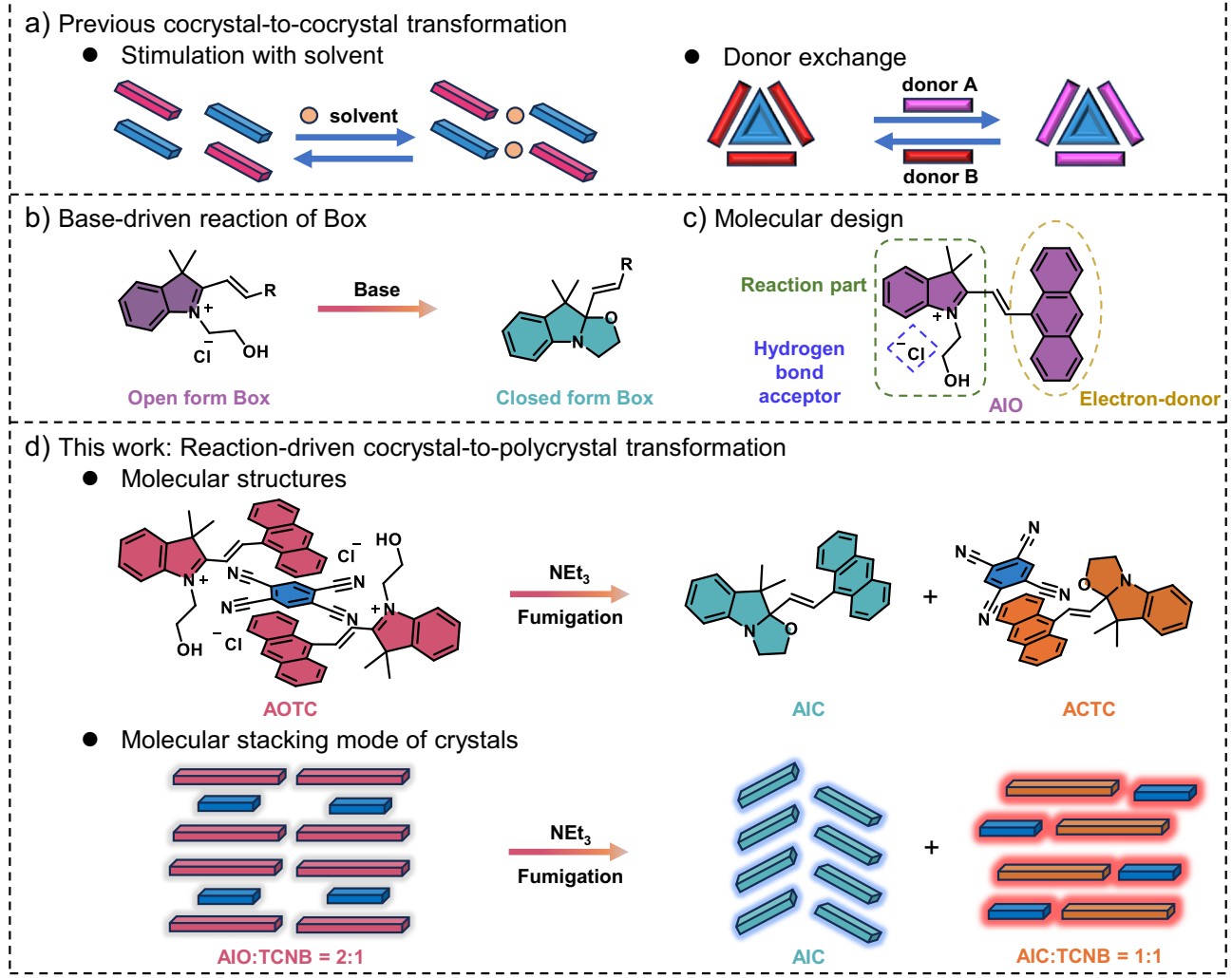

**Fig. 1 | Strategies for cocrystal transformations. a** Previously reported cocrystal-to-cocrystal transformations. **b** Molecular switching of indolino[2,1-b]oxazolidine (box) derivatives enables conversion from the open form to the closed form upon treatment with the base. **c** Molecular design of AIO. **d** Schematic illustration and molecular stacking mode of the cocrystal-to-polycrystal transformation driven by NEt₃.

cocrystal-to-polycrystal transformation (Fig. 1d). The color, fluorescence, and appearance exhibit notable differences after exposure to NEt₃, providing a desirable platform to visualize the phenomenon of cocrystal transformation. Furthermore, the transformation was successfully utilized for information encryption. These systems demonstrate a rare cocrystal-to-polycrystal transformation accompanied by a change in cocrystal stoichiometric ratio. This transformation paves the way for crystal engineering applicable to functional materials.

## Results

TCNB, with its four electron-deficient units, commonly interacts with electron-rich molecules to form charge-transfer cocrystals. For this study, the 9-anthracene-substituted indolino[2,1-b]oxazolidine (Box) in the open form (AIO) was chosen as the electron donor. Dark red cuboid cocrystals of AOTC, which do not exhibit fluorescence under UV light, were obtained by slow evaporation of a mixed acetonitrile solution of AIO and TCNB (Fig. 2a). The cocrystal structure was subsequently confirmed by X-ray single-crystal diffraction (XRD) analysis (Supplementary Table 1), which revealed a 2:1 molecular ratio of AIO to TCNB. Detailed structural information is provided in Supplementary Figs. 1–2. Powder X-ray diffraction (PXRD) analysis showed a new set of diffraction peaks for the AOTC cocrystal, distinct from those of the single-component crystals (TCNB or AIO), further confirming the formation of AOTC (Supplementary Fig. 3). The Fourier transform infrared (FTIR) spectra (Supplementary Fig. 4) of the AOTC cocrystal contained features from both AIO and TCNB crystals. Furthermore, the thermogravimetric analysis demonstrated a smooth baseline,

indicating that the AOTC cocrystal is stable below 220 °C (Supplementary Fig. 5). These results collectively suggest that the AOTC cocrystal was successfully achieved through the co-assembly of TCNB and AIO.

Multiple intermolecular interactions were observed in the AOTC cocrystal between AIO and TCNB molecules, as shown in Fig. 2a. The chloride ions form hydrogen bonds with the hydroxyl groups and the aryl C–H groups in TCNB, with O–H···Cl⁻ and C–H···Cl⁻ distances of 2.29 and 2.47 Å, respectively[53]. Each TCNB molecule interacts with two chloride ions, corresponding to a stoichiometric ratio of 2:1 (Fig. 2a). Moreover, each TCNB molecule stacks face-to-face with two anthracene groups, with D–A distances of 3.61 Å (Fig. 2b). The electron density maps also revealed that TCNB exhibited low electronic density (deep-blue color), which could form intermolecular D–A interactions with the high electron density AIO molecule (red and yellow color) (Supplementary Fig. 6). The hydrogen bonds and D–A interactions are directly evident in the independent gradient model based on Hirshfeld partition (IGMH) analysis[54], which may collectively contribute to the cocrystal formation through a mixed-stacking mode (Fig. 2c). As shown in Fig. 2d and Supplementary Figs. 7–10, the Hirshfeld surface analysis and two-dimensional (2D) fingerprint plots indicate that the percentage of intermolecular N···H interactions increased from 0% in AIO to 16.7% in AOTC. This interaction between the TCNB and AIO molecules (as shown by the brown dashed line in Fig. 2b) helps to maintain the crystallinity during crystal transformation[55].

Since base treatment can induce the open-to-closed conversion of Box, the transformation of AOTC cocrystals was investigated.

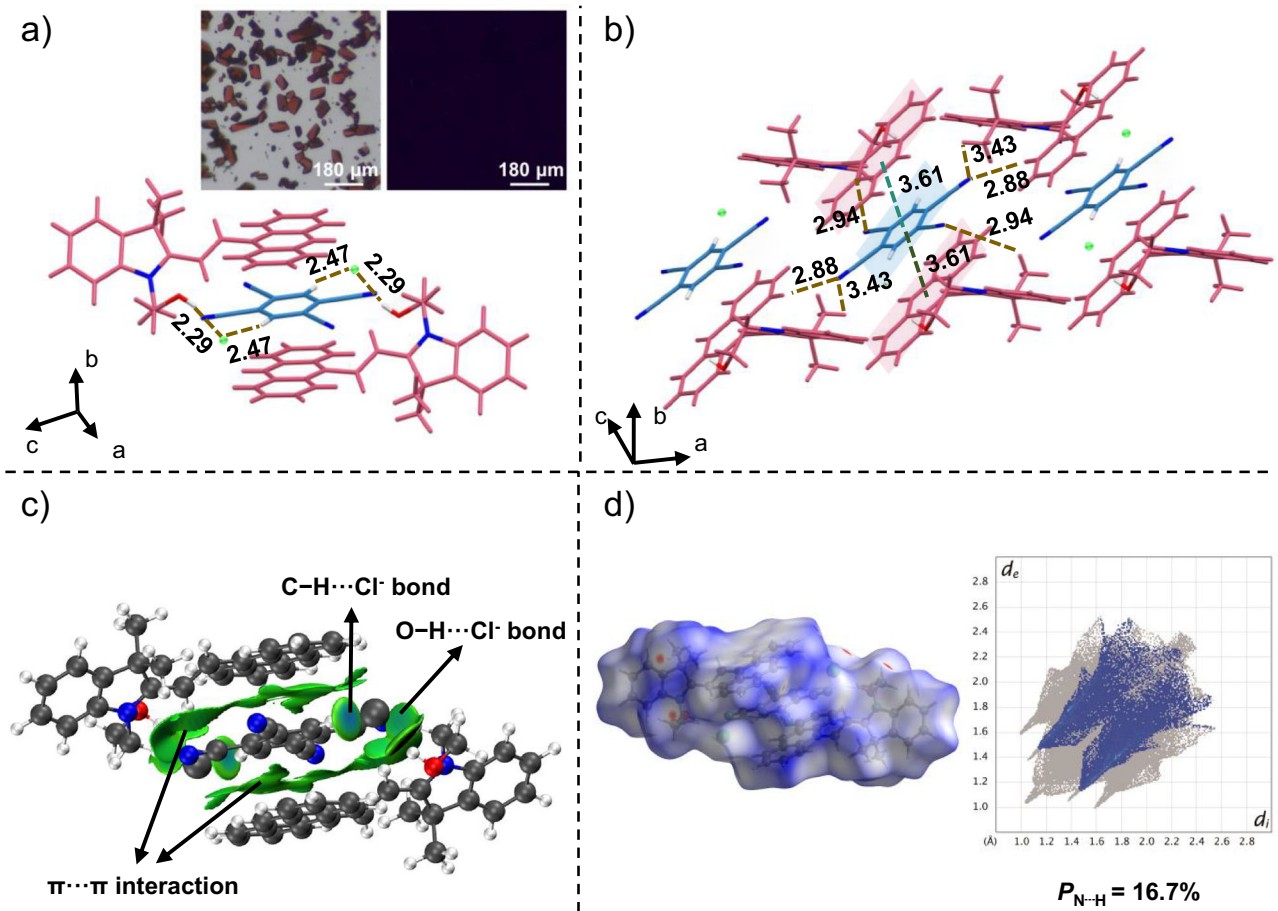

**Fig. 2 | The crystal structure of AOTC cocrystal. a** Optical microscope images of the AOTC cocrystal taken under room light (left) and under 365 nm UV irradiation (right), and the crystal structure of AOTC. **b** The π···π interactions and C–H···N interactions in the crystal structure of AOTC. **c** IGMH analysis of the non-covalent interactions in AOTC. **d** Hirshfeld surfaces and decomposed fingerprint plots showing N···H interactions in AOTC, with the proportions of N···H interaction ($P_{N···H}$) relative to total intermolecular interactions also indicated.

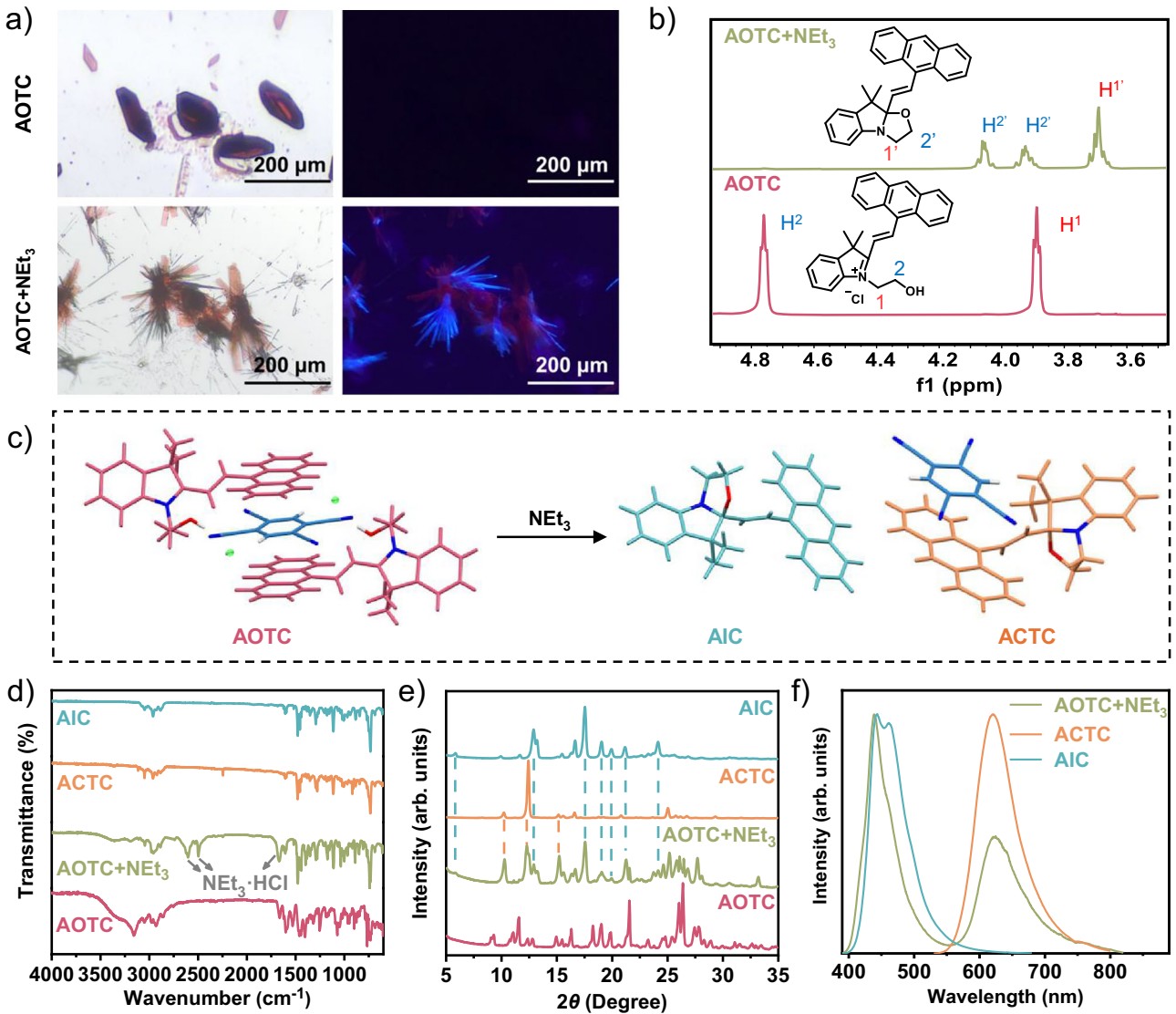

**Fig. 3 | Cocrystal-to-polycrystal transformation. a** Optical microscope images showing the cocrystal-to-polycrystal transformation under room light (left) and 365 nm UV irradiation (right). Top: the initial AOTC cocrystals; bottom: the transformed AOTC+NEt₃ crystals. **b** Partial ¹H NMR spectra of the initial AOTC cocrystal and the transformed AOTC+NEt₃ crystal (DMSO-$d_6$, 500 MHz). **c** Molecular structures of the single crystals of AOTC, AIC, and ACTC. **d** FTIR spectra of AIC, ACTC, AOTC+NEt₃, and AOTC. **e** PXRD patterns of AIC, ACTC, AOTC+NEt₃, and AOTC. **f** Solid-state fluorescence spectra of AOTC+NEt₃, ACTC, and AIC.

Exposure of the AOTC cocrystals to triethylamine vapor at room temperature (Supplementary Fig. 11), the dark red cuboid AOTC cocrystals gradually transformed into orange-striped and yellow rod-shaped crystals. This transformation was observed in situ and over various durations (Fig. 3a and Supplementary Fig. 12) under a microscope. Fluorescence microscopy further revealed that the cocrystal transitioned from non-emissive to exhibiting blue and red emission. The distinct changes in morphology and fluorescence of the newly formed polycrystal prompted a deeper structural analysis. The initial detection of the transformation was achieved through ¹H NMR spectra by comparing the original AOTC with the AOTC exposed to triethylamine vapor (referred to as AOTC+NEt₃). As shown in Fig. 3b, the differences in chemical shifts and peak splitting indicate a structural transformation from the open to the closed form. ¹H NMR spectrum of the AOTC+NEt₃ also indicated the simultaneous formation of Et₃N·HCl, as evidenced by proton signals at 10.20, 3.05, and 1.20 ppm (Supplementary Fig. 13). Subsequent XRD analysis confirmed that the yellow rod-shaped crystal is a single-component AIC crystal of the closed form Box, while the orange-striped crystal is a cocrystal of AIC and TCNB (named ACTC), indicating a transition from cocrystal to

polycrystal (Fig. 3c, Supplementary Figs. 14 and 15 and Supplementary Table 2). Notably, the ratio of AIC and TCNB in ACTC is 1:1, differing from that in AOTC (2:1), highlighting the distinct configuration in the cocrystals formed by TCNB with the reactants and products. This transformation underscores that the changes in the donor-acceptor ratios drive the transition from a single cocrystal to polycrystals.

The recrystalliztion of AOTC cocrystal in liquid triethylamine was also investigated. This process leads to the partial dissolution of the cocrystal without the formation of either single-crystal AIC or the cocrystal ACTC (Supplementary Fig. 16). In addition, exposure of the ACTC cocrystals to acids such as CF₃COOH, AcOH, or HCl vapors resulted in the dissolution or fragmentation of the cocrystals (Supplementary Fig. 17). Therefore, the transformation of the cocrystals is irreversible.

To further elucidate the cocrystal-to-polycrystal transformation, AIC crystals and ACTC cocrystals were also successfully prepared by slow evaporation of solutions in acetonitrile (Supplementary Figs. 18 and 19), allowing for a comparative analysis of their properties. Initially, FTIR spectra were carried out to investigate the compositional changes during the cocrystal structural transformation (Fig. 3d). These

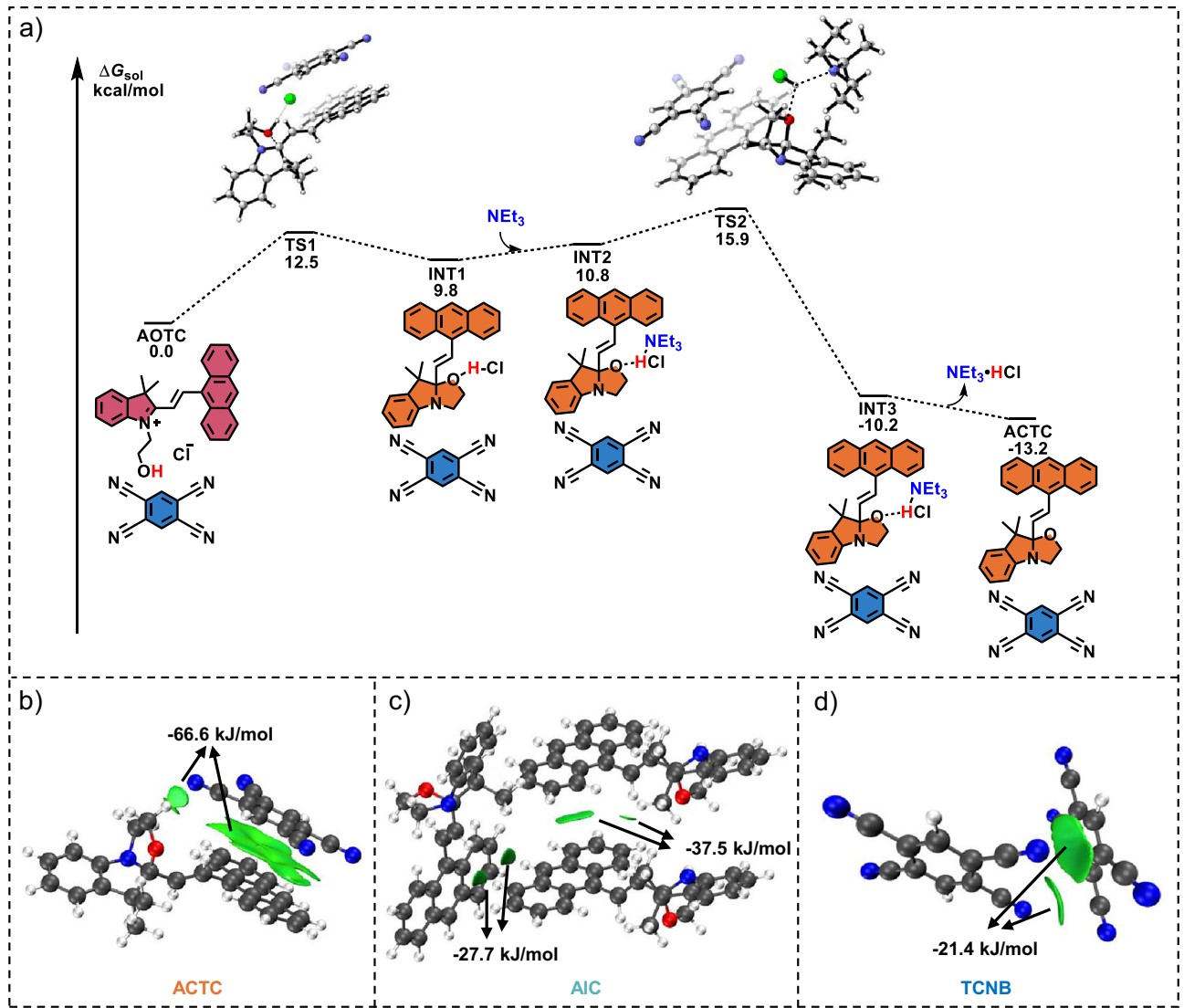

**Fig. 4 | Research on the mechanism of cocrystal transformation. a** Energy profile for the cyclization reaction of AOTC at the CAM-B3LYP/6-31 + G* level with SMD solvent model (triethylamine). For computational simplicity, a Box and a TCNB molecule were selected to participate in the process. IGMH analysis of the non-covalent interactions in **b** ACTC, **c** AIC, and **d** TCNB. The interaction energies were calculated and analyzed using EDA-FF methods.

spectra clearly show the disappearance of the hydroxyl group in AOTC +NEt$_3$, as evidenced by the absence of the characteristic FTIR peak around 3160 cm$^{-1}$. Additionally, the presence of FTIR peaks around 2496, 2603, and 1658 cm$^{-1}$ in AOTC+NEt$_3$ indicates the formation of NEt$_3$·HCl. To further determine the structural changes before and after fuming with NEt$_3$, PXRD characterizations were conducted. The PXRD patterns of AOTC+NEt$_3$ matched a combination of AIC and ACTC (Fig. 3e), along with traces of powdered NEt$_3$·HCl (Supplementary Fig. 20), providing additional evidence of the transformation from AOTC cocrystal to AIC crystal and ACTC cocrystal.

Solid-state absorption and fluorescence spectra of both reactant and product crystals were recorded to characterize their photophysical properties (Supplementary Fig. 21). Due to the oxazolidine ring opening/closure in the Box moiety and the D−A interactions between the anthracene and TCNB moieties, the reactant and product crystals exhibit distinctive optical properties. Specifically, the solid-state UV-Vis absorption spectra for AOTC, AOTC+NEt$_3$, ACTC, and AIC reveal broad absorption bands. The AOTC+NEt$_3$ shows a blue-shifted absorption band compared to AOTC, peaking at ~570 nm, which resembles the absorption spectra of ACTC mixed with AIC. The AIC crystal and the ACTC cocrystal (both in the closed form) exhibit broad

absorption bands ranging from 250 to 410 nm and 250 to 570 nm, respectively. The red-shifted absorption of the ACTC cocrystal, compared to the AIC crystal, results from the D−A interactions between the electron donor and acceptor[19].

The AOTC crystals were non-emissive, consistent with previous findings for open-form Box derivatives, likely due to electron transfer between the excitation fluorophore and the electron-deficient group[56,57]. However, fluorescence was generated when treated with NEt$_3$. The solid-state fluorescence spectra of AOTC+NEt$_3$ exhibit blue luminescence with a peak at 439 nm and red luminescence at 622 nm ($\lambda_{ex}$ = 380 nm) (Fig. 3f). These spectra were generally similar to those of the directly prepared AIC and ACTC crystals (Fig. 3f and Supplementary Fig. 22), with the slight differences compared to pure AIC likely due to the influence of NEt$_3$·HCl (Supplementary Fig. 23). These indicate that the transition from AOTC cocrystals to AIC and ACTC cocrystals leads to significant changes in luminescence properties which provide an effective method for monitoring the cocrystal transformation process.

To elucidate the mechanism by which NEt$_3$ promotes cocrystal transformation, theoretical calculations were employed to reveal the cyclization reaction mechanism of AOTC (Fig. 4a)[58]. Initially, the

oxygen atom attacks the iminium ion through a low-energy barrier transition state (TS1, 12.5 kcal/mol) to form intermediate INT1. Subsequently, the final product ACTC is produced through NEt$_3$-assisted hydrochloric acid removal via transition state TS2 (15.9 kcal/mol). DFT calculations show that the NEt$_3$-assisted cyclization reaction of AOTC is both kinetically and thermodynamically favorable, with a lower energy barrier compared to its individual components (16.9 kcal/mol) (Supplementary Fig. 24), thereby facilitating the cocrystal transformation.

The formation of the ACTC cocrystal rather than single crystals of AIC or TCNB can be attributed to the strong intermolecular interactions between AIC and TCNB. In the ACTC cocrystal, AIC and TCNB molecules stack alternately, with anthracene units in AIC engaging in face-to-face D–A interactions with TCNB at 3.79 Å (Supplementary Fig. 25). These strong D–A interactions were further confirmed by Hirshfeld surfaces (Supplementary Fig. 26), 2D fingerprint plots (Supplementary Figs. 27 and 28), ground-state electronic potential calculations (Supplementary Fig. 29), and Frontier molecular orbitals (Supplementary Fig. 30). Quantitative analysis using IGMH and EDA-FF methods[59] reveals that the total interaction energy between AIC and TCNB is −66.6 kJ/mol due to D–A interaction and C–H···N hydrogen bonds (Fig. 4b). The energy is significantly higher than the π–π interaction energies between AIC-AIC (−37.5 kJ/mol or −27.7 kJ/mol, Fig. 4c) and TCNB-TCNB (−21.4 kJ/mol, Fig. 4d). Thus, the stronger D–A and hydrogen bonding interactions between AIC and TCNB drive the formation of the ACTC cocrystal instead of single-component crystals.

Based on the experimental and theoretical calculations, a plausible process for the cocrystal transformation can be proposed. Upon exposure to NEt$_3$ vapor, molecules on the surface of the AOTC cocrystal gradually dissolve and diffuse into the surrounding microsolution, causing surface corrosion (Supplementary Fig. 31). Concurrently, AIO molecules in the AOTC cocrystal react with triethylamine to form closed-box products (AIC molecules). As the NEt$_3$ solution evaporates, free AIC molecules cocrystallize with neighboring TCNB molecules to form ACTC microcrystals in a 1:1 molar ratio, driven by stronger D–A interactions and C–H···N hydrogen bonds. The change in molecular structure alters the non-covalent intermolecular interactions, leading to a shift in cocrystal stoichiometry. Excess AIC molecules then aggregate to form AIC single-component crystals. This process is similar to recrystallization in the microsolution formed by triethylamine vapor. Initially, new small polycrystals form on the surface of the AOTC cocrystals and continue to grow after the formation of new cyclization products (Supplementary Figs. 12 and 32). Other bases, including ammonia (NH$_3$), *N, N*-Diisopropylethylamine, and sodium carbonate solution, failed to induce the transformation due to they could not generate sufficient vapor or did not create an effective microsolution on the crystal surface (Supplementary Fig. 33). Thus, NEt$_3$ vapor facilitates the transition from cocrystal to polycrystals by promoting partial dissolution of AOTC cocrystals and participating in the cyclization reaction, enabling continuous crystal transformations while maintaining overall crystallinity.

The significant color change observed during the cocrystal-to-polycrystal transformation inspired us to explore its applications in information encryption. As shown in Fig. 5a, a floral pattern was created with the AOTC solution for the flowers and the AIO solution for the leaves. Upon spraying NEt$_3$ on the painting, the leaf pattern disappeared instantly, while the flowers remained visible. In another example, two brushes dipped in AOTC and AIO solutions were used to draw different parts of the number "6789" on paper (left panel of Fig. 5b). Although the pattern appeared secure to the naked eye, exposing the paper to NEt$_3$ vapor revealed the true number "5134," separating it from the misinformation. Additionally, the letters "ing" written with the AOTC solution were hidden when the surrounding surface was sprayed with the AIO solution (Fig. 5c), but were revealed upon exposure to NEt$_3$ vapor. Subsequent spraying with hydrochloric acid caused the information to disappear again. This color switching could be repeated

about four times with alternating base-acid cycles, demonstrating the stability and reversible responsiveness of the samples (Fig. 5c and Supplementary Fig. 34). Notably, UV-visible titration experiments revealed that the interaction between the two components in solution is relatively weak (Supplementary Fig. 35). This effectively rules out the possibility that the observed phenomenon arises from complex formation between TCNB and AIO/AIC under soluble conditions, especially considering that all subsequent operations in our study were conducted after drying the encrypted paper samples. Therefore, the information encryption behavior can be attributed to the properties of the crystalline solids formed on the paper. These findings suggest that the cocrystal-to-polycrystal transformation systems hold potential for information encryption applications.

## Discussion

In this work, we have demonstrated a cocrystal-to-polycrystal transformation induced by NEt$_3$ vapor. The AOTC cocrystal was successfully prepared with 9-anthracene-substituted indolino-oxazolidine and TCNB through hydrogen bonding and D–A interactions. The obtained AOTC cocrystal, with a 2:1 stoichiometric ratio, exhibits no fluorescence. Upon exposure to NEt$_3$ vapor, this non-emissive AOTC cocrystal transforms into a red-emissive ACTC cocrystal and a blue-emissive AIC single-component crystal. Remarkably, the transformation from cocrystal to polycrystal preserves crystallinity throughout the process, as confirmed by single-crystal X-ray diffraction analysis. The transition from the 2:1 AOTC cocrystal to the 1:1 ACTC cocrystal results in a transition from cocrystal to multiple polycrystals. In this process, NEt$_3$ vapor plays a critical role: it partially dissolves the AOTC cocrystal and participates in the cyclization reaction, facilitating continuous crystal transformations while maintaining crystallinity. Theoretical calculations reveal that the cyclization reaction has a low-energy barrier of 15.9 kcal/mol, and the formation of the ACTC cocrystal is driven by cooperative D–A intermolecular interactions and C–H···N hydrogen bond interactions. Additionally, reliable information encryption was achieved based on the base-responsive properties of these cocrystals and crystals. This work presents an advanced strategy for precisely controlling cocrystal transformation while retaining the crystalline state through a vapor-induced cyclization reaction. These findings offer a model for crystal transformation that could inspire further exploration into chemical reactions capable of inducing crystal transformations, potentially advancing fields such as crystalline molecular machines, materials science, and supramolecular chemistry.

## Methohds
### Materials
All starting materials, reagents and anhydrous solvents were purchased from commercial suppliers (Aldrich, Alfa, TCI, etc.) and used as supplied unless otherwise stated. The TCNB (97%) was purchased from Sigma-Aldrich.

### Synthesis of (*E*)-9a-(2-(anthracen-9-yl)vinyl)-9,9-dimethyl-2,3,9,9a-tetrahydrooxazolo[3,2-*a*]indole (AIC)[60,61].
The indolinooxazolidin (1.0 mmol, 0.2 g) and 9-anthraldehyde (1.0 mmol, 0.2 g) were added to 1 g of silica, and the heterogeneous mixture was homogenized by the addition of a minimal amount (1–2 mL) of DCM. The solvent was removed under reduced pressure at ambient temperature. Then, the powder was heated to 100 °C and continuously stirred for 2 h under normal atmospheric conditions. The residue was purified by column chromatography on silica gel (petroleum ether: ethyl acetate = 10:1) to afford the desired products as a yellow solid (0.294 g, 75% yield). Mp: 154–155 °C; $^1$H NMR (500 MHz, DMSO-$d_6$) δ 8.58 (s, 1H), 8.24 (d, J = 8.8 Hz, 2H), 8.12 (d, J = 8.4 Hz, 2H), 7.63–7.52 (m, 5H), 7.21–7.15 (m, 2H), 6.99–6.90 (m, 2H), 6.07 (d, J = 16.2 Hz, 1H), 4.11–4.01 (m, 1H), 3.96–3.88 (m, 1H), 3.75–3.63 (m, 2H), 1.52 (s, 3H), 1.22 (s, 3H); $^{13}$C NMR (125 MHz, DMSO-$d_6$) δ 150.64, 139.11, 134.43, 131.67, 130.95, 128.82,

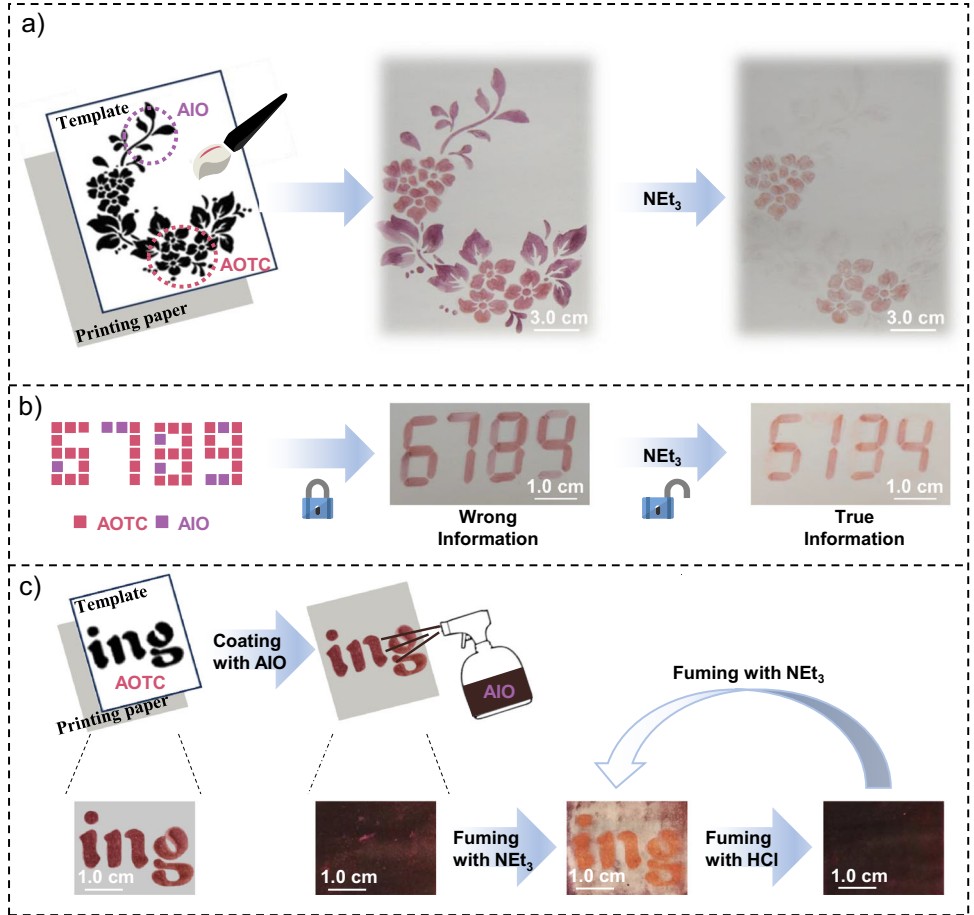

**Fig. 5 | Applications in information encryption. a** NEt₃-induced stimulus-responsive process of a pattern created with AOTC and AIO. **b** Information encryption platform using AOTC and AIO, revealing encoded information upon exposure to NEt₃. **c** Demonstration of the reversibility of the information storage model made with AOTC and AIO upon sequential exposure to NEt₃ and HCl.

128.71, 128.08, 127.55, 126.33, 126.05, 125.38, 125.19, 122.28, 121.37, 112.17, 109.38, 63.51, 49.99, 47.32, 28.40, 20.43; IR (neat): 3048, 2961, 1606, 1595, 1477, 1455, 1441, 1352, 1292, 1278, 1113, 1021, 1005, 884, 768, 749, 732; HRMS (ESI) calcd. for $C_{28}H_{25}NO$ [M + H]⁺: 392.2009, found: 392.2000.

**Synthesis of (*E*)-2-(2-(anthracen-9-yl)vinyl)-1-(2-hydroxyethyl)-3,3-dimethyl-3*H*-indol-1-ium chloride (AIO).** The product AIC (1.0 mmol, 0.392 g) was dissolved in DCM (20 mL), and the aqueous HCl solution (1 M, 10 mL) was added. The reaction was stirred at room temperature for 30 min. The resulting solution was extracted with DCM (15.0 mL × 3). The combined organic layers were dried over anhydrous $Na_2SO_4$, filtered, and concentrated in vacuo. The residue was recrystallized in DCM and hexane to afford a product as a dark red solid (0.405 g, 95% yield). Mp: 155–156 °C; ¹H NMR (500 MHz, DMSO-$d_6$) δ 9.43 (d, *J* = 16.5 Hz, 1H), 8.86 (s, 1H), 8.36 (d, *J* = 8.6 Hz, 2H), 8.23 (d, *J* = 8.3 Hz, 2H), 8.07–8.04 (m, 1H), 8.02–7.99 (m, 1H), 7.74–7.63 (m, 7H), 4.76 (t, *J* = 4.8 Hz, 2H), 3.89 (t, *J* = 4.8 Hz, 2H), 2.05 (s, 6H); ¹³C NMR (125 MHz, DMSO-$d_6$) δ 183.09, 150.24, 144.08, 140.94, 130.85, 130.63, 129.80, 129.42, 129.17, 129.12, 127.64, 125.96, 125.23, 123.22, 123.13, 115.96, 58.36, 52.88, 49.87, 25.70; IR (neat): 3232, 3184, 3052, 2927, 1623, 1587, 1527, 1476, 1458, 1448, 1314, 1251, 1188, 1075, 957, 764, 742, 734; HRMS (ESI) calcd. for $C_{28}H_{26}ClNO$ [M-Cl]⁺: 392.2009, found: 392.2004.

**Crystal growth methods**
For cocrystal AOTC: AOTC cocrystals were prepared using the slow evaporation method. AIO and TCNB were mixed with the ratio of 2:1 in acetonitrile. Then, the solutions were allowed to evaporate slowly at

room temperature. After 3–5 days, dark red block crystals were obtained.

For cocrystal ACTC: ACTC cocrystals were prepared using the slow evaporation method. AIC and TCNB were mixed in 1:1 ratio in acetonitrile. Then, the solutions were allowed to evaporate slowly at room temperature. After 3–5 days, orange-striped cocrystals were obtained.

AIO was prepared by slow solvent evaporation in acetonitrile.
AIC was prepared by slow solvent evaporation in acetonitrile.

**Characterizations**
Nuclear magnetic resonance spectra of ¹H and ¹³C were tested on Bruker 500 MHz Spectrometer (¹H: 500 MHz; ¹³C: 125 MHz). Chemical shifts (δ) are given in ppm relative to TMS. The residual solvent signals were used as references, and the chemical shifts were converted to the TMS scale (($CD_3$)$_2$SO: δ $H$ = 2.50 ppm, δ C = 39.52 ppm). UV-vis absorption spectra were recorded on a Shimadzu UV-2700 UV-visible spectrophotometer with an ISR-2600 Integrating Sphere Attachment to measure the diffuse reflectance of the solid materials. The photoluminescence spectra were recorded on a HORIBA Fluorolog-QM fluorescence spectrofluorometer. The absolute fluorescence quantum yield was measured by a calibrated integrating sphere (Labsphere). The high-resolution electron spray ionization mass spectra (HR ESI-MS) were performed on an Agilent (Santa Clara, CA, USA) ESI-TOF mass spectrometer (6224) and micrOTOF mass spectrometer. Single-crystal X-ray diffraction data were collected on an Xcalibur E diffractometer with graphite monochromated Cu-K$_\alpha$ radiation (λ = 1.54184 Å), and crystal structures were solved with Olex2. PXRD

pattern was collected on a Rigaku Ultima IV diffractometer using Cu-K$_\alpha$ radiation at 35 kV and 25 mA in the 2$\theta$ angle range of 5–50° using a step size of 0.02° and at a scanning speed of 10° min$^{-1}$. The fluorescent images of the crystals were taken by an OLYMPUS CKX53 microscope. Scanning electron microscopy (SEM) images of the crystals were obtained using an S4800 (Hitachi Ltd.) with an accelerating voltage of 3.0-10.0 k. Thermogravimetric analyses were carried out on a Perkin Elmer STA8000 thermal analyzer under a nitrogen atmosphere with a heating rate of 10 °C min$^{-1}$. The FTIR spectra were obtained using SHIMADZU TRT racer-100.

## Data availability

The authors declare that all the data supporting the findings of this manuscript are available within the manuscript and Supplementary Information files and available from the corresponding authors upon request. The X-ray crystallographic coordinates for structures reported in this study have been deposited at the Cambridge Crystallographic Data Center (CCDC), under deposition numbers 2343124 (AIC), 2343125 (AIO), 2343126 (ACTC), and 2343127 (AOTC). These data can be obtained free of charge from The Cambridge Crystallographic Data Center via www.ccdc.cam.ac.uk/data_request/cif. Atomic coordinates of the optimized computational models source data are present. Source data are provided with this paper.

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

## Acknowledgements

This work was supported by the National Natural Science Foundation of China (no. 22371076 to L.X.), the Program of Shanghai Outstanding Academic Leaders (no. 21XD1421200 to L.X.), "Shuguang Program" supported by the Shanghai Education Development Foundation and Shanghai Municipal Education Commission (no. 22SG22 to L.X.), Science and Technology Commission of Shanghai Municipality (Nos. 22JC1403900 and 21520710200 to H.B.Y.), and the Fundamental Research Funds for the Central Universities (East China Normal University to L.X.).

## Author contributions

L.Z., L.H., L.X., and B.Z.T. conceived the idea and designed the experiments. L.Z. carried out all the experiments, including the preparations, characterization of the materials, and data analysis. X.Z. and Y.N. conducted the SXRD experiments and analyzed the data. W.D. and H.B.Y. supervised the work. L.Z. conducted the theoretical calculations. L.Z., L.H., L.X., and B.Z.T. revised the manuscript and put forward explanations for the mechanism. All authors have approved the final version of the manuscript.

## Competing interests

The authors declare no competing interest.
