## [Transparent Peer Review file · Nature Communications]

Triethylamine Vapor-Induced Cyclization Reaction in Cocrystals Leading to Cocrystal-to-Polycrystal Transformation

Corresponding Author: Professor Ben Zhong Tang

Version 0:

Reviewer comments:

Reviewer #1

(Remarks to the Author)

In this work, Tang and co-authors have introduced a triethylamine vapor-induced strategy to transform the cocrystal AOTC into single-crystal AIC and cocrystal ACTC. This innovative approach integrates chemical reactions with crystal transformation, presenting a novel mechanism to achieve crystal transitions. Especially, achieving this transformation while preserving the crystalline state through vapor represents a truly novel concept. The combination of experimental and theoretical analyses highlights that the cocrystal-to-polycrystal transformation is primarily driven by the cyclization reaction, as well as the synergistic effects of intermolecular D–A interactions and C–HN hydrogen bonding. This work represents a sophisticated strategy for controlling cocrystal transformations while preserving the crystalline state, using a vapor-induced cyclization reaction. Overall, the study of vapor-induced cyclization reactions leading to cocrystal-to-polycrystal transformations will be of broad interest to researchers in crystalline molecular machines, materials science, and supramolecular chemistry, and that it merits publication in Nature Communications, provided the following minor issues are adequately addressed:

1. The authors have chosen triethylamine as the vapor to induce the cocrystal transformation. Why was triethylamine specifically selected? Have the authors considered or explored other alkaline vapors for this process? Providing a rationale or comparison with alternative vapors would enhance the discussion.
2. Herein, the term “cocrystals” mainly refers to “molecular cocrystals” or “organic cocrystals”. In the introduction, when describing the “cocrystals”, the related review in this field (Chem. Eur. J. 2015, 21, 4880) can be included as reference.
3. Is the cocrystal-to-polycrystal transformation process reversible? Given the known acid-base reversibility of the indolino[2,1-b]oxazolidine (Box) system, can acidic vapor induce the transformation of the closed-form 9-anthracene-substituted indolino-oxazolidine cocrystal (ACTC) back into AOTC cocrystals? Clarifying this aspect would provide valuable insights into the transformation mechanism.
4. In Figure 4c, the green dot represents weak interactions between AIC-AIC molecules. However, the corresponding interaction energy does not appear to have been calculated. Does this affect the comparison of interaction energies between different cocrystals? Including this calculation or explaining its impact would strengthen the analysis.
5. Both cocrystals and crystal transformation are hot topics in chemistry and materials. To arouse a broad interest from readership in this field, some strongly related works on cocrystals materials and their crystal transformation (CCS Chemistry, 2023, 5, 2866; Chem. Eur. J. 2013, 19, 8213; J. Mater. Chem. C 2016, 4, 2527) could be included as references.
6. Several formatting issues need to be addressed for consistency and clarity. For instance, "Figs. 1d" on page 4 could be corrected to "Fig. 1d", the phrase "high resolution electron spray ionization mass spectra" in the characterization section could be revised to "high-resolution electron spray ionization mass spectra".

Reviewer #2

(Remarks to the Author)

This study describes the conversion of AOTC (cocrystal of 9-anthracene-substituted indoline-oxazolidine, Box and 1,2,4,5-tetracyanobenzene, TCNB) into a mixed crystal of AIC (closed form of AOTC) and ACTC (a cocrystal of AIC and TCNB). This transformation is triggered by triethylamine vapor and involves the ring-closure of 9-anthracene-substituted indoline-oxazolidine (Box) compounds. The term “cocrystal-to-polycrystal transformation” is included in the title, but since this

phenomenon involves dissolution and recrystallization on the crystal surface due to high concentrations of triethylamine, it is more accurate to refer to it as recrystallization rather than vapor-triggered crystal-to-crystal transformation. By exposing AOTC crystals to triethylamine (Figure S11), the crystal's external shape changes significantly, and as also the authors comment, there is no doubt that this is a phenomenon caused by partial dissolution in triethylamine. Although the author used triethylamine vapor for transportation, this is not vapor-induced transformation. This phenomenon is quite different from the more interesting vapor-driven phase transition reported in the past, such as the pioneering work by Kato et al. (Kato, M.; Omura, A.; Toshikawa, A.; Kishi, S.; Sugimoto, Y. *Angew. Chem., Int. Ed.* 2002, 41, 3183–3185).

Regarding the “information processing” in the second half, there have been many reports of demonstrations that claim to have “processed information” in experiments like this, but they are not actually usable so far and are not very meaningful, so they do not serve to increase the value of this research.

Essentially, there have been many reports of recrystallization which can produce many different donor-acceptor crystals from other crystals, like this result, and throughout this research, it seems that it does not reach the level of originality that would be equivalent to Nature Comm at least at the current stage. In order to improve the quality of this research, I would like to propose the following revisions at the very least.

1. The cyclization product is chiral, but there is no discussion of this in relation to the crystal. Is the product a racemic pair, or is there a fractional crystallization of enantiomers occurring?
2. Regarding the DFT calculations, the information on the basis sets and methods used needs to be shown in the main text. This is because the results of the calculations will differ greatly depending on these choices. Also, since the reaction probably occurs in triethylamine, a comment on this is also needed. It would be good to add calculations that incorporate triethylamine as a solvent effect.
3. Where is Et₃NHCl located in the products? This should be clearly indicated and discussed.
4. What happens if other bases, such as NH₃, are used?
5. What happens if the product is recrystallized using triethylamine as the solvent?
6. The relevant research on vapo-chromism should be correctly cited. In addition to the above paper, Kato, M.; Ito, H.; Hasegawa, M.; Ishii, K. *Chem. Eur. J.* 2019, 25, 5105–5112, etc.
7. The title should include “recrystallization” to show the phenomena more precisely. For example, the title can be “Triethylamine-Induced Cyclization Reaction of Cocrystals Leading to Cocrystal-to-Polycrystal Surface Recrystallization.”

Reviewer #3

(Remarks to the Author)

The manuscript entitled “Triethylamine Vapor-Induced Cyclization Reaction in Cocrystals Leading to Cocrystal-to-Polycrystal Transformation”, submitted by Ben Zhong Tang and co-workers, presents the modulation of crystal phases in materials prepared via the crystallization of an anthracene-indolino-oxazolidine (AIO) dyad and tetracyanobenzene (TCNB), a well-known electron-poor compound. More specifically, this study reports the “cocrystal-to-polycrystal” transformation of the material, along with the variation of its optical properties when exposed to triethylamine (NEt₃) as an organic base. In addition to extensive optical characterizations, the authors substantiate their assumptions with DFT calculations and appropriate analyses of the harvested crystallographic data.

Impressed by this commendable work and the rather uncommon “Cocrystal-to-Polycrystal Transformation” properties that could interest a broad readership, I fully support the publication of this manuscript in “Nature Communications” journal. However, I have identified several points, outlined below, that should be addressed to enhance the manuscript's impact and clarity prior to its final publication.

Major points:

1/ The exposure to NEt₃ is known to induce the cyclization of oxazolidine rings in various BOX derivatives in their open forms, typically resulting in the formation of triethylamine hydrochloride (NEt₃-HCl) as a side product. Surprisingly, in this manuscript, the presence of NEt₃-HCl is only sporadically and inconsistently mentioned in the context of the “Cocrystal-to-Polycrystal Transformation.” The formation of this high-melting-point white solid should be discussed more thoroughly. Was the presence of such white crystals formation is noticed in the reaction medium (on reaction vessel) after exposing AOTC to NEt₃ vapor? Indeed, if I understand correctly, the XRD data of the resulting crystals did not reveal the presence of NEt₃-HCl.

To address this, I recommend providing the full ¹H NMR spectrum of AOTC crystals treated with NEt₃ and dissolved in DMSO (this could be simply included in the supplementary materials). This data would confirm whether NEt₃ or NEt₃-HCl is integrated into the crystal structure or mixed with the material. Furthermore, accurate integration of the ¹H NMR signals would confirm the 1:1 stoichiometry of the assembly with TCNB and clarify whether the transformation from AOTC to ACTC is complete or partial. This point, which is currently unclear, should be also explicitly addressed

2/ Reverse Transformation (Polycrystal-to-Cocrystal):

The cyclization of BOX derivatives with organic bases is well-documented, as is their reverse transformation to the open form upon treatment with acids (e.g., HCl). It is surprising that the manuscript does not mention attempts to achieve the reverse transformation on solid state, i.e., converting polycrystals back to cocrystals, by exposing the material to acids such as HCl or TFA.

This point is relevant, particularly as the authors briefly touch on the use of this reversible transformation for information encryption (lines 273–276). Even if such experiments were unsuccessful, this should be explicitly mentioned and discussed.

3/ Behavior in Solution for Information Encryption:

Since the information encryption experiments were conducted from solution of AOTC and AIO crystals, the potential for complexation between TCNB and AIO/AICN in solution must be investigated. I suggest conducting UV-Visible titrations of AIO and AICN solutions with TCNB in acetonitrile, as well as titrations of AIO/TCNB and AICN/TCNB mixtures with some base and some acid, respectively.

Minor points:

1/ Sentence Clarity and Figure References (lines 96–99):

The sentence “This process intriguingly led... cocrystal-to-polycrystal transformation (Figs. 1d)” is difficult to follow. I recommend rephrasing it and adding a link to Fig. 3a. Additionally, consider replacing the optical microscope images in Fig. 3a with those from Fig. S29 (0h and 48h), which more clearly demonstrate the crystal transformation (AOTC to ACTC) and the growth of new crystals (AIC) on it. Similarly, the reference to Fig. S12 (line 153) is useless and should be replaced by figure S29 more suitable to illustrate the “in-situ” observation of the crystal to crystal transformation.

2/ Solid-State Absorption Spectra (line 190):

The link to Fig. S18 is missing. Furthermore, the spectra presented are unusual, showing nearly continuous absorption without distinct maxima. The authors should explain this observation, especially since similar compounds exhibit a strong and distinct band around 250 nm in solution exhibits in solution a very intense and distinct band around 250nm (almost 6 times higher than in UVA range). In this context, the method used to record the spectra (e.g., film or diffuse reflectance) should be described more explicitly in proper section of the manuscript in place of only reporting the brand mark of the used spectrophotometer.

3/ Stating on line 241, The authors hypothesize that NEt_3 solvation drives the “Cocrystal-to-Polycrystal Transformation”. To support this, they could attempt to grow AIC and ACTC crystals via slow evaporation of AOTC solutions in pure NEt_3 or NEt_3 /acetonitrile mixtures

4/ The manuscript suggests that fluorescence changes from dual to single emission for AIC (lines 205-209) due to $\text{NEt}_3\cdot\text{HCl}$, although $\text{NEt}_3\cdot\text{HCl}$ does not appear to be present in the crystals. The basis for assigning to AIC a dual emission at 443 and 461 nm should be clarified, as these could instead arise from vibronic structuring or aggregate formation. Additionally, the emissive wavelength used for lifetime measurements in Fig. S19c should be specified.

5/ I was surprised to read line 309 in material section that “all solvents were dried over Na”. Please Clarify which solvents were dried using such method. Indeed, they report the used of DCM as solvent to prepare AIC but the addition of reactive metals such sodium to chlorinated solvents can result in strong safety disappointment. Similarly, for acetonitrile (used for crystallization) is known to react with alkali metals.

Version 1:

Reviewer comments:

Reviewer #1

(Remarks to the Author)

In my view, the authors have answered all the questions and revised related points, and thus this revised work can be published as it is.

Reviewer #2

(Remarks to the Author)

I believe the issues I pointed out have been addressed. However, I still have some concerns regarding the originality of the work. I will leave the final decision to the editor's judgment.

Reviewer #3

(Remarks to the Author)

The manuscript entitled "Triethylamine Vapor-Induced Cyclization Reaction in Cocrystals Leading to Cocrystal-to-Polycrystal Transformation", submitted by Ben Zhong Tang and co-workers, is a revised version of a manuscript submitted a few months ago by the same authors, which I previously reviewed.

In this version, the authors have addressed almost all of my concerns described in my previous report, and as a consequence, I support its publication in Nature Communications. Nevertheless, some minor points (listed below) need to be corrected prior to publication, without necessarily requiring a further review process.

Concerning comment 3.1 regarding the triethylamine hydrochloride salt, I am pleased that the authors have followed my recommendation to include the full ^1H NMR spectra. However, the modification of the text with the insertion of the sentence:

"¹H NMR spectrum of AOTC+NEt₃ also displayed a signal corresponding to equivalent Et₃N·HCl, with proton signals at 10.20, 1.20, and 3.05 ppm, revealing the formation of the salt mixed with the crystals (Fig. S13)."

could lead readers to believe that only one triethylamine molecule is sufficient to convert one AOTC unit (involving two BOX units and one TCNB). The ¹H NMR signal of an ethylenic proton of the BOX derivative (a doublet around 6 ppm, see ref. 50) is a very convenient way to determine the open/closed state ratio and the cis/trans conformation of numerous BOX derivatives. Based on its integration (0.96) and that of the triethylamine hydrochloride salt signal at 10.2 ppm (0.99), we can conclude a 1:1 ratio between the BOX unit and HNEt₃. Additionally, the 2:1 ratio between BOX and TCNB is also confirmed, as the integration of the singlet at 9.03 ppm (0.84) corresponds to the two aromatic protons of TCNB.

Concerning comment 3.3 regarding Information Encryption, I am pleased that the authors have performed the requested complementary experiments in solution. Despite their assumption about the formation of AOTC crystals on paper, I continue to support the inclusion of these data (in the supporting information, with a brief mention in the main text). This is motivated by

-First, no experimental data (PXRD, TEM images, etc.) demonstrate the effective crystallization of AOTC in a cellulosic medium.

-Second, the absence of such unusual chromic phenomenon in solution seems to support the crystallization hypothesis while ruling out the possibility of complex formation in solution.

Concerning comment 3.5, the authors did not fully address my concern. While I completely agree with them that a strong variation could be observed between the UV-Visible spectra in solution and in the solid state. However, this continuum in the UVA region could also result from the experimental setup. In this context, my recommendation was to specify the method used to acquire the UV-Visible spectra of the different solid materials. Consequently, the insertion of the sentence: "UV-Vis absorption spectra were recorded on a Shimadzu UV-Visible spectrophotometer and measured under standard testing conditions with solid" (line 373) is not sufficient. The model (such as example SolidSpec-3700i ??) and the type of performed measurement (was it transmittance or diffuse reflectance of the solid material? Or was it transmittance of the powder confined between two quartz/glass plates?) should be specified

Concerning comment 3.7, I continue to support the removal of the term dual emission from this manuscript, as it is not fully appropriate in this context.

As far as I am concerned, the term dual emission generally refers to the presence of two radiative de-excitation pathways, which are explained in terms of excited states of different nature (J. Phys. Chem. C 2013, 117, 35, 18154), such as in the case of luminescence from excimers, twisted intramolecular charge transfer (TICT) states, etc.

By contrast, describing the emission band as having a vibronic structure seems more accurate in this case, as it implies the presence of only one type of excited state. As clearly demonstrated by the authors, the observation of the vibronic structure in the AIC emission band—attributable to the anthracene moiety—is strongly influenced by the surrounding medium. Generally, anthracene excimer luminescence is observed at much longer wavelengths.

made.

Response Letter to Reviewers

Response to the Comments from Reviewer #1:

In this work, Tang and co-authors have introduced a triethylamine vapor-induced strategy to transform the cocrystal AOTC into single-crystal AIC and cocrystal ACTC. This innovative approach integrates chemical reactions with crystal transformation, presenting a novel mechanism to achieve crystal transitions. Especially, achieving this transformation while preserving the crystalline state through vapor represents a truly novel concept. The combination of experimental and theoretical analyses highlights that the cocrystal-to-polycrystal transformation is primarily driven by the cyclization reaction, as well as the synergistic effects of intermolecular D-A interactions and CH-N hydrogen bonding. This work represents a sophisticated strategy for controlling cocrystal transformations while preserving the crystalline state, using a vapor-induced cyclization reaction. Overall, the study of vapor-induced cyclization reactions leading to cocrystal-to-polycrystal transformations will be of broad interest to researchers in crystalline molecular machines, materials science, and supramolecular chemistry, and that it merits publication in Nature Communications, provided the following minor issues are adequately addressed:

Reply: We sincerely appreciate the reviewer's positive evaluation and insightful comments on our work. In response to the reviewer's valuable suggestions, we have carefully revised the manuscript to improve its clarity and rigor. We hope that the reviewer finds our responses and revisions satisfactory.

Comment 1.1: The authors have chosen triethylamine as the vapor to induce the cocrystal transformation. Why was triethylamine specifically selected? Have the authors considered or explored other alkaline vapors for this process? Providing a rationale or comparison with alternative vapors would enhance the discussion.

Reply: We appreciate the reviewer's insightful question.

Triethylamine was selected due to its low boiling point (89°C), which allows it to generate vapor at room temperature. This vapor creates a microsolution environment that facilitates the oxazolidine ring closure in the AOTC cocrystal, thereby driving the cocrystal transformation. Moreover, triethylamine has been previously reported to promote the cyclization of oxazolidine derivatives (*J. Am. Chem. Soc.* **2019**, *141*, 19151–19160), further supporting its suitability for this reaction. In contrast, most other organic bases have higher boiling points and do not readily form vapor under ambient

conditions, limiting their effectiveness in this transformation. Moreover, as shown in Fig. S33, we investigated several alternative organic and inorganic bases, including ammonia (NH₃), *N,N*-diisopropylethylamine, and sodium carbonate solution. However, none of these were able to induce the cocrystal-to-polycrystal transformation, as they either failed to generate sufficient vapor or did not create an effective microsolution on the crystal surface. To enhance the discussion, we have incorporated this rationale, along with comparative analysis and experimental data, into the revised manuscript and Supporting Information.

Fig. S33 (a) Optical microscope images of the AOTC cocrystal before and after exposure to NH₃. (b) Optical microscope images of the AOTC cocrystal following exposure to *N,N*-diisopropylethylamine. (c) Optical microscope images of the AOTC cocrystal after immersion in a sodium carbonate solution.

Comment 1.2: Herein, the term “cocrystals” mainly refers to “molecular cocrystals” or “organic cocrystals”. In the introduction, when describing the “cocrystals”, the related review in this field (Chem. Eur. J. 2015, 21, 4880) can be included as reference.

Reply: We sincerely appreciate the reviewer’s constructive suggestion. In accordance with the recommendation, we have cited the relevant review (Chem. Eur. J. 2015, 21, 4880) in the manuscript to clarify the definition of “organic cocrystals” in this context.

Comment 1.3: Is the cocrystal-to-polycrystal transformation process reversible? Given the known acid-base reversibility of the indolino[2,1-b]oxazolidine (Box) system, can acidic vapor induce the transformation of the closed-form 9-anthracene-substituted indolino-oxazolidine cocrystal (ACTC) back into AOTC cocrystals? Clarifying this aspect would provide valuable insights into the transformation mechanism.

Reply: We appreciate the reviewer’s insightful comments. Regrettably, the cocrystal-to-polycrystal transformation process observed in our study is not reversible. As shown in Fig. S15, exposure of the ACTC cocrystals to CF₃COOH, AcOH, and HCl vapors at room temperature did not result in the formation of new crystals. In the CF₃COOH and AcOH systems, the cocrystals gradually dissolved in the vapor-condensed solution. When ACTC cocrystals were exposed to HCl vapor, the closed-form substrate

transitioned to the open-form, causing the cocrystals to fragment within 30 minutes. These observations indicate that, despite the known acid-base reversibility of the indolino[2,1-b]oxazolidine (Box) system, the specific transformation between the ACTC and AOTC cocrystals is not reversible. Moving forward, we plan to investigate other cocrystal systems and explore alternative reversible reactions to enable the reversible transformation of cocrystals.

Fig. S15 Optical microscope images of ACTC cocrystal after exposure to (a) CF_3COOH , (b) AcOH , and (c) HCl .

Comment 1.4: In Figure 4c, the green dot represents weak interactions between AIC-AIC molecules. However, the corresponding interaction energy does not appear to have been calculated. Does this affect the comparison of interaction energies between different cocrystals? Including this calculation or explaining its impact would strengthen the analysis.

Reply: We thank the reviewer for highlighting this important point. The weak intermolecular interactions between AIC-AIC molecules at the bottom of Figure 4c were indeed calculated and previously identified. However, the green dot that was omitted in the figure actually represents an intramolecular interaction, which should not have been included in the interaction energy calculation. To avoid any confusion, we have revised Figure 4c to display only the relevant weak intermolecular interactions associated with the target molecules. These revisions have been highlighted with a yellow background in the manuscript. We believe these adjustments will enhance the clarity and accuracy of the analysis.

Fig. 4 a) Energy profile for the cyclization reaction of AOTC at the CAM-B3LYP/6-31+G* level with SMD solvent model (triethylamine). For computational simplicity, a Box and a TCNB molecule were selected to participate in the process. IGMH analysis of the non-covalent interactions in b) ACTC, c) AIC, and d) TCNB. The interaction energies were calculated and analyzed using EDA-FF methods.

Comment 1.5: Both cocrystals and crystal transformation are hot topics in chemistry and materials. To arouse a broad interest from readership in this field, some strongly related works on cocrystals materials and their crystal transformation (CCS Chemistry, 2023, 5, 2866; Chem. Eur. J. 2013, 19, 8213; J. Mater. Chem. C 2016, 4, 2527) could be included as references.

Reply: We appreciate the reviewer's suggestion. In response, we have incorporated the suggested references into the manuscript. These references provide valuable context and will help broaden the appeal of our work to readers interested in cocrystals and crystal transformation.

Comment 1.6: Several formatting issues need to be addressed for consistency and

clarity. For instance, "Figs. 1d" on page 4 could be corrected to "Fig. 1d", the phrase "high resolution electron spray ionization mass spectra" in the characterization section could be revised to "high-resolution electron spray ionization mass spectra".

Reply: We appreciate the reviewer's careful attention to detail. In response to the suggestions, we have corrected the formatting issues, including changing "Figs. 1d" to "Fig. 1d" and revising the phrase "high resolution electron spray ionization mass spectra" to "high-resolution electron spray ionization mass spectra" in the characterization section.

Response to the Comments from Reviewer #2:

This study describes the conversion of AOTC (cocrystal of 9-anthracene-substituted indoline-oxazolidine, Box and 1,2,4,5-tetracyanobenzene, TCNB) into a mixed crystal of AIC (closed form of AOTC) and ACTC (a cocrystal of AIC and TCNB). This transformation is triggered by triethylamine vapor and involves the ring-closure of 9-anthracene-substituted indoline-oxazolidine (Box) compounds. The term “cocrystal-to-polycrystal transformation” is included in the title, but since this phenomenon involves dissolution and recrystallization on the crystal surface due to high concentrations of triethylamine, it is more accurate to refer to it as recrystallization rather than vapor-triggered crystal-to-crystal transformation. By exposing AOTC crystals to triethylamine (Figure S11), the crystal's external shape changes significantly, and as also the authors comment, there is no doubt that this is a phenomenon caused by partial dissolution in triethylamine. Although the author used triethylamine vapor for transportation, this is not vapor-induced transformation. This phenomenon is quite different from the more interesting vapor-driven phase transition reported in the past, such as the pioneering work by Kato et al. (Kato, M.; Omura, A.; Toshikawa, A.; Kishi, S.; Sugimoto, Y. *Angew. Chem., Int. Ed.* 2002, 41, 3183–3185).

Regarding the “information processing” in the second half, there have been many reports of demonstrations that claim to have “processed information” in experiments like this, but they are not actually usable so far and are not very meaningful, so they do not serve to increase the value of this research.

Essentially, there have been many reports of recrystallization which can produce many different donor-acceptor crystals from other crystals, like this result, and throughout this research, it seems that it does not reach the level of originality that would be equivalent to Nature Comm at least at the current stage. In order to improve the quality of this research, I would like to propose the following revisions at the very least.

Reply: We sincerely thank the reviewer for the insightful comments and detailed feedback. We have comprehensively addressed all the issues raised subsequently, and we believe that the revisions, along with the additional experiments conducted, effectively resolve the concerns and enhance the originality of our work. We are confident that these changes have significantly improved the quality and clarity of the manuscript, and we look forward to refine our work based on your valuable suggestions.

Comment 2.1: The cyclization product is chiral, but there is no discussion of this in relation to the crystal. Is the product a racemic pair, or is there a fractional crystallization

of enantiomers occurring?

Reply: We appreciate the reviewer's insightful comment. The AIC and ACTC crystals are indeed racemic. As illustrated in Figures S17 and S19, the R/S chirality of the oxazolidine molecules within the cocrystals has been clarified. We have updated the figures in the revised Supporting Information to provide a more detailed depiction of the chirality in the crystal structures. The modified figures now replace the original ones in the Supporting Information.

Fig. S17. Molecular packing modes of AIC crystal perpendicular to the (a) a-axis, (b) b-axis, and (c) c-axis.

Fig. S19. Molecular packing modes of ACTC crystal perpendicular to the (a) a-axis, (b) b-axis, and (c) c-axis.

Comment 2.2: Regarding the DFT calculations, the information on the basis sets and methods used needs to be shown in the main text. This is because the results of the calculations will differ greatly depending on these choices. Also, since the reaction probably occurs in triethylamine, a comment on this is also needed. It would be good to add calculations that incorporate triethylamine as a solvent effect.

Reply: We thank the reviewer for these insightful comments.

In response to the request, we have now included detailed information regarding the basis sets and methods used for the DFT calculations in the main text. The relevant revisions have been highlighted with a yellow background for clarity.

Following the reviewer's suggestion, we have incorporated the SMD solvent model to account for the solvent effects, specifically considering triethylamine as the solvent for the DFT calculations of the cyclization reaction. The results from these updated calculations are consistent with our previous findings. The revised energy profiles, incorporating the solvent effect, are now shown in Figures 4a and S24. We believe these updates will enhance the accuracy and clarity of the manuscript.

Fig. 4 a) Energy profile for the cyclization reaction of AOTC at the CAM-B3LYP/6-31+G* level with SMD solvent model (triethylamine).

Fig. S24 DFT computed free energy changes for the cyclization reaction of AIO at the CAM-B3LYP/6-31+G* level with SMD solvent model (triethylamine).

Comment 2.3: Where is Et_3NHCl located in the products? This should be clearly indicated and discussed.

Reply: Thank you for raising this important point. We appreciate the opportunity to clarify the presence and location of Et_3NHCl in the products.

The FTIR spectra and PXRD patterns of AOTC+NEt₃ (Figures 3d and 3e) provide clear evidence of the presence of Et₃NHCl in the products. These spectra indicate the characteristic bands corresponding to the Et₃NHCl salt.

The full ¹H NMR spectra of the transformed AOTC+NEt₃ crystal further confirm the presence of Et₃NHCl. A distinctive peak at 10.20 ppm is attributed to the proton of the product salt, with a quantity matching that of AIC. Additionally, the triplet and quartet peaks observed at 1.20 and 3.05 ppm, respectively, are assigned to the protons of triethylamine part. The quantity of triethylamine present is slightly higher than theoretical expectations, due to trace amounts of free triethylamine molecules remaining in the sample. To further clarify this point, we have added the full ¹H NMR spectra to the Supporting Information (Figure S13).

In the optical microscope images, Et₃NHCl is not easily distinguishable due to its colorless, transparent nature and small crystal size. Additionally, it may be obscured by the AIC and ACTC crystals. We speculate that the fluorescent non-emissive solid outlined by the red dashed line in Figure 3a corresponds to Et₃NHCl, although its non-emissive properties and small size prevent it from being clearly identified by XRD.

We hope this additional information provides clarity on the location of Et₃NHCl in the products.

Fig. S13 ¹H NMR spectrum of the transformed AOTC+NEt₃ crystal (samples were

dissolved in DMSO-*d*₆).

Fig. 3a Optical microscope images of the transformed AOTC+NEt₃ crystals. Possible Et₃NHCl are circled by red dashed line.

Comment 2.4: What happens if other bases, such as NH₃, are used?

Reply: Thank you for your insightful question.

As shown in Figure S33, we have investigated several other organic and inorganic bases, including NH₃, *N,N*-Diisopropylethylamine, and sodium carbonate solution, to explore their potential for inducing the cocrystal-to-polycrystal transformation. However, none of these bases were successful in inducing the transformation, as they were unable to form vapor or microsolution on the surface of the crystals. Specifically, when AOTC cocrystals were exposed to NH₃ vapor, the transparent dark red cocrystals turned into an opaque solid without crystal structure and no new crystals formed (Figure S33a).

We chose triethylamine for this process due to its low boiling point (89°C), which allows it to generate solvent vapor at room temperature. This vapor induces the oxazolidine ring closure in the AOTC cocrystal, providing a microsolution environment conducive to cocrystal transformation. Additionally, triethylamine has been previously reported to facilitate the cyclization reaction of oxazolidine derivatives (*J. Am. Chem. Soc.* **2019**, *141*, 19151–19160). In contrast, most other organic bases have relatively higher boiling points and cannot generate vapor at room temperature, limiting their ability to induce the transformation.

Fig. S33 (a) Optical microscope images of the AOTC cocrystal before and after exposure to NH₃. (b) Optical microscope images of the AOTC cocrystal following exposure to *N,N*-diisopropylethylamine. (c) Optical microscope images of the AOTC

cocrystal after immersion in a sodium carbonate solution.

Comment 2.5: What happens if the product is recrystallized using triethylamine as the solvent?

Reply: Thank you for your thoughtful suggestion.

We have explored the recrystallization of AOTC cocrystals using liquid triethylamine as the solvent. This process leads to the partial dissolution of the cocrystal without the formation of single-crystal AIC or the cocrystal ACTC (Fig. R1a/S14). Under fluorescence microscopy, colorless needle-shaped crystals show no fluorescence, unlike the blue and red emission observed in vapor-driven recrystallization, indicating the absence of AIC or ACTC crystals (Fig. R1b). To investigate further, we collected the supernatant, concentrated it in vacuo, and dissolved the residue in CD₃Cl for ¹H NMR analysis (Fig. R1c). The ¹H NMR spectrum shows a mole fraction of AIC:TCNB of 5:1, which differs from the 2:1 ratio in AOTC cocrystals. This result suggests that AIC and TCNB have different solubilities in triethylamine, with AIC being more soluble than TCNB. As a result, recrystallization in liquid triethylamine only produces colorless needle-shaped TCNB crystals (Fig. R1a/S14).

Additionally, we attempted to place AOTC cocrystals in liquid triethylamine and slowly evaporate the solvent (Fig. R1d). This led to the formation of colorless, transparent microcrystals emitting blue fluorescence, alongside non-emissive TCNB or NEt₃·HCl crystals, indicating that AIC molecules dissolved in triethylamine after the cyclization reaction and recrystallized as the solvent evaporated. The solid-state fluorescence spectrum of the product shows blue luminescence at 439 nm without any red luminescence (Fig. R1e), confirming that only single-crystal AIC was generated, without the formation of the cocrystal ACTC.

In summary, the experimental results show that only triethylamine vapor can trigger the conversion of AOTC cocrystals into polycrystals, whereas recrystallization in liquid triethylamine does not yield the two crystals. This indicates that the transformation from cocrystal to polycrystal is not a recrystallization process; instead, vapor-induced transformation plays a crucial role. We have included the relevant content and figures (Fig. S14) in the revised manuscript and Supporting Information.

Fig. R1 (a/S14) Optical microscope images of AOTC cocrystals in liquid triethylamine at different time intervals. (b) Optical microscope images (left) and fluorescence microscope images (right) of AOTC cocrystals immersed in triethylamine. (c) Partial ^1H NMR spectra of the supernatant fluid. (d) Optical microscope images (left) and fluorescence microscope images (right) of AOTC cocrystals placed in liquid triethylamine, followed by slow evaporation of the triethylamine solution. (e) Solid-state fluorescence spectra of the product obtained by the slow evaporation of the triethylamine solution.

Comment 2.6: The relevant research on vapochromism should be correctly cited. In addition to the above paper, Kato, M.; Ito, H.; Hasegawa, M.; Ishii, K. *Chem. Eur. J.* 2019, 25, 5105–5112, etc.

Reply: Thank you for your valuable suggestion. In response, we have updated the manuscript to correctly cite the relevant research on vapochromism, including the work by Kato et al. (*Chem. Eur. J.* 2019, 25, 5105–5112) and other related studies.

Comment 2.7: The title should include “recrystallization” to show the phenomena more precisely. For example, the title can be “Triethylamine-Induced Cyclization Reaction of Cocrystals Leading to Cocrystal-to-Polycrystal Surface Recrystallization.”

Reply: Thank you for your valuable suggestion. We agree that the phenomenon observed in our study differs significantly from vapor-driven phase transitions reported in previous research, which we have addressed in the introduction of the manuscript. However, we believe the term “recrystallization” may not be entirely suitable for our study. In the literature, “recrystallization” is generally defined as a process where the crystal structure changes without any alteration to the chemical structure (*Materials Science and Engineering A* 1997, 238, 219–274; *Progress in Materials Science* 2014,

60, 130–207). Additionally, we have explored the recrystallization of AOTC cocrystals in liquid triethylamine and allowed the solvent to evaporate slowly. However, these processes did not lead to the formation of single-crystal AIC or the cocrystal ACTC (Fig. R1), indicating that the transformation from cocrystal to polycrystal is not a recrystallization process. Therefore, we prefer not to use "recrystallization" or "surface recrystallization" in the title.

We feel that the process described in our manuscript represents a type of crystal “transformation”. The term “transformation” has been widely used in the literature to describe changes in both crystal structures and chemical compositions induced by vapors or solvents (*J. Am. Chem. Soc.* **2020**, *142*, 7265–7269; *J. Am. Chem. Soc.* **2023**, *145*, 1855–1865; *Nanoscale* **2019**, *11*, 8692; *Chem. Commun.* **2023**, *59*, 11413). Therefore, we consider "transformation" to be a more accurate descriptor for the phenomenon observed in our study.

In response to your suggestion, we have further clarified in the manuscript that the observed phenomenon bears similarity to recrystallization in the microsolution formed by triethylamine vapor. This additional description helps to provide a more precise depiction of the experimental phenomenon.

Response to the Comments from Reviewer 3:

The manuscript entitled "Triethylamine Vapor-Induced Cyclization Reaction in Cocrystals Leading to Cocrystal-to-Polycrystal Transformation", submitted by Ben Zhong Tang and co-workers, presents the modulation of crystal phases in materials prepared via the crystallization of an anthracene-indolino-oxazolidine (AIO) dyad and tetracyanobenzene (TCNB), a well-known electron-poor compound. More specifically, this study reports the "cocrystal-to-polycrystal" transformation of the material, along with the variation of its optical properties when exposed to triethylamine (NEt_3) as an organic base. In addition to extensive optical characterizations, the authors substantiate their assumptions with DFT calculations and appropriate analyses of the harvested crystallographic data.

Impressed by this commendable work and the rather uncommon "Cocrystal-to-Polycrystal Transformation" properties that could interest a broad readership, I fully support the publication of this manuscript in "Nature Communications" journal. However, I have identified several points, outlined below, that should be addressed to enhance the manuscript's impact and clarity prior to its final publication.

Reply: We sincerely thank the reviewer for the positive and encouraging comments on our manuscript. In response to the valuable suggestions and concerns raised, we have made careful revisions and provided detailed explanations for each point. We hope that the revisions and clarifications we have made will address the reviewer's concerns and further enhance the clarity and impact of the manuscript.

Major points:

Comment 3.1: The exposure to NEt_3 is known to induce the cyclization of oxazolidine rings in various BOX derivatives in their open forms, typically resulting in the formation of triethylamine hydrochloride ($\text{NEt}_3 \cdot \text{HCl}$) as a side product. Surprisingly, in this manuscript, the presence of $\text{NEt}_3 \cdot \text{HCl}$ is only sporadically and inconsistently mentioned in the context of the "Cocrystal-to-Polycrystal Transformation." The formation of this high-melting-point white solid should be discussed more thoroughly. Was the presence of such white crystals formation is noticed in the reaction medium (on reaction vessel) after exposing AOTC to NEt_3 vapor? Indeed, If I understand correctly, the XRD data of the resulting crystals did not reveal the presence of $\text{NEt}_3 \cdot \text{HCl}$. To address this, I recommend providing the full ^1H NMR spectrum of AOTC crystals treated with NEt_3 and dissolved in DMSO (this could be simply included in the supplementary materials). This data would confirm whether NEt_3 or $\text{NEt}_3 \cdot \text{HCl}$ is

integrated into the crystal structure or mixed with the material. Furthermore, accurate integration of the ^1H NMR signals would confirm the 1:1 stoichiometry of the assembly with TCNB and clarify whether the transformation from AOTC to ACTC is complete or partial. This point, which is currently unclear, should be also explicitly addressed

Reply: Thank you for your insightful comments and valuable suggestions. We appreciate your careful review of the manuscript. Below are our detailed responses to address your concerns:

The presence of triethylamine hydrochloride ($\text{NEt}_3\cdot\text{HCl}$) in optical microscope images is difficult to identify due to its colorless, transparent nature and small crystal size. Additionally, it may be obscured by the AIC and ACTC crystals. We hypothesize that the non-emissive solid circled by a red dashed line in Fig. 3a represents $\text{NEt}_3\cdot\text{HCl}$, though its non-emissive behavior and small size prevent it from being directly observed or tested by XRD.

The XRD patterns of the resulting crystals (AIC and ACTC) did not reveal the presence of $\text{NEt}_3\cdot\text{HCl}$ because this compound did not co-crystallize with AIC or ACTC during the transformation process.

To substantiate the formation of $\text{NEt}_3\cdot\text{HCl}$, we have included FTIR spectra and PXRD patterns of the products ($\text{AOTC}+\text{NEt}_3$) in Figs. 3d and 3e. These analyses confirm the presence of $\text{NEt}_3\cdot\text{HCl}$ as a byproduct in the transformation.

In response to your suggestion, we have included the full ^1H NMR spectrum of the transformed $\text{AOTC}+\text{NEt}_3$ crystal in the Supporting Information (Fig. S13). As shown in Fig. S13, the characteristic peak at 10.20 ppm corresponds to the $\text{NEt}_3\cdot\text{HCl}$ salt, with an amount equivalent to AIC. The triplet and quartet peaks at 1.20 and 3.05 ppm correspond to the triethylamine (NEt_3) part. The quantity of NEt_3 is slightly higher than theoretical expectations, likely due to the presence of small amounts of free NEt_3 molecules. Furthermore, the spectrum confirms that the transformation from AOTC to ACTC is complete.

We have revised the manuscript to include a more thorough discussion of the $\text{NEt}_3\cdot\text{HCl}$ formation and its role in the transformation process. We believe these additional data and clarifications address your concerns and improve the clarity of our findings.

Fig. 3a Optical microscope images of the transformed AOTC+NEt₃ crystals. Possible Et₃NHCl are circled by red dashed line.

Fig. S13 ¹H NMR spectrum of the transformed AOTC+NEt₃ crystal (samples were dissolved in DMSO-d₆).

Comment 3.2: Reverse Transformation (Polycrystal-to-Cocrystal):

The cyclization of BOX derivatives with organic bases is well-documented, as is their reverse transformation to the open form upon treatment with acids (e.g., HCl). It is surprising that the manuscript does not mention attempts to achieve the reverse transformation on solid state, i.e., converting polycrystals back to cocrystals, by exposing the material to acids such as HCl or TFA.

This point is relevant, particularly as the authors briefly touch on the use of this reversible transformation for information encryption (lines 273–276). Even if such experiments were unsuccessful, this should be explicitly mentioned and discussed.

Reply: Thank you for your insightful suggestion and for raising this important point. Here are our detailed responses:

Unfortunately, the cocrystal-to-polycrystal transformation is irreversible. As shown in Fig. S15, exposure of ACTC cocrystals to CF_3COOH , AcOH, and HCl vapors at room temperature did not lead to the formation of new crystals. In the CF_3COOH and AcOH systems, the cocrystals gradually dissolved in the vapor-condensed solution. When ACTC cocrystals were exposed to HCl, the closed-form substrate transformed into the open form, causing the cocrystals to fragment within 30 minutes.

Although the acid treatment did not result in the reversible formation of cocrystals, it did cause significant color changes in the solids due to the molecular structural transformation. This property makes it suitable for potential applications in reversible information encryption.

We have included the discussion of the irreversibility of the acid-induced transformation in the revised manuscript, as suggested. We acknowledge that further exploration of other cocrystals and reversible reactions will be essential for enabling the reversible transformation of cocrystals, and we plan to investigate this in future studies.

Fig. S15 Optical microscope images of ACTC cocrystal after exposure to (a) CF_3COOH , (b) AcOH, and (c) HCl.

Comment 3.3: Behavior in Solution for Information Encryption:

Since the information encryption experiments were conducted from solution of AOTC and AIO crystals, the potential for complexation between TCNB and AIO/AICN in solution must be investigated.

I suggest conducting UV-Visible titrations of AIO and AICN solutions with TCNB in acetonitrile, as well as titrations of AIO/TCNB and AIC/TCNB mixtures with some base and some acid, respectively.

Reply: Thank you for your valuable suggestion. We appreciate your insightful comments, and we have carefully addressed the points raised. Here is a detailed response:

We would like to clarify that the information encryption experiments were performed using microcrystals, not in solution. The patterns for encryption were created using

AOTC and AIO solutions, with the AOTC and AIC crystals forming on the paper after the solvent had evaporated. This patterning technique is a well-established method for information encryption, as demonstrated in previous studies (*Angew. Chem. Int. Ed.* **2024**, *63*, e202318497; *J. Am. Chem. Soc.* **2021**, *143*, 9468–9477; *J. Am. Chem. Soc.* **2021**, *143*, 1553–1561).

In response to your suggestion, we performed UV-visible titrations of AIO and AIC solutions with TCNB in acetonitrile. However, the UV-visible spectra showed minimal changes (Fig. R2a and R2b), indicating weak charge-transfer (CT) interactions in solution. Due to the weak nature of these interactions, Job's plot method could not reliably determine the binding ratio between AIO/AIC and TCNB in solution (Fig. R2c and R2d).

We also carried out titrations of AIO/TCNB and AIC/TCNB mixtures with a base (NEt_3) and an acid (HCl), as suggested. The results, shown in Fig. R3a and R3b, demonstrate clear acid-base response behavior. Adding NEt_3 to the AIO/TCNB solution led to a reduction in absorption at 508 nm, while adding HCl to the AIC/TCNB solution resulted in a significant absorption band at 508 nm. These results confirm the acid-base response behavior of the two solutions, which is consistent with the observed dramatic color changes in the AIO/AIC systems.

Fig. R2 (a) UV-vis spectra of AIO in acetonitrile solution (0.1 mM) with varying

amounts of TCNB (0–3.0 equiv.). (b) UV-vis spectra of AIC in acetonitrile solution (0.1 mM) with varying amounts of TCNB (0–3.0 equiv.). (c) UV-vis spectra of the AIO/TCNB complex formed by mixing different molar ratios of AIO and TCNB in acetonitrile solution (0.1 mM). (d) UV-vis spectra of the AIC/TCNB complex formed by mixing different molar ratios of AIC and TCNB in acetonitrile solution (0.1 mM).

Fig. R3 (a) UV-vis spectra of the AIO/TCNB complex in acetonitrile solution (0.1 mM) with varying amounts of NEt₃ (0–1 equiv.). (b) UV-vis spectra of the AIC/TCNB complex in acetonitrile solution (0.1 mM) with varying amounts of HCl (0–1 equiv.).

Minor points:

Comment 3.4: Sentence Clarity and Figure References (lines 96–99):

The sentence “This process intriguingly led... cocrystal-to-polycrystal transformation (Figs. 1d)” is difficult to follow. I recommend rephrasing it and adding a link to Fig. 3a. Additionally, consider replacing the optical microscope images in Fig. 3a with those from Fig. S29 (0 h and 48 h), which more clearly demonstrate the crystal transformation (AOTC to ACTC) and the growth of new crystals (AIC) on it. Similarly, the reference to Fig. S12 (line 153) is useless and should be replaced by figure S29 more suitable to illustrate the “in-situ” observation of the crystal to crystal transformation.

Reply: We would like to thank the reviewer for pointing out these important issues.

In response to the suggestion, we have rephrased the sentence for clarity: “*The change in molecular structure intriguingly led to the transformation of the AOTC cocrystal into single crystals containing only the closed form Box (AIC), as well as cocrystals of AIC and TCNB (named ACTC), demonstrating a cocrystal-to-polycrystal transformation (Figs. 1d and 3a).*” The revised sentence is highlighted with a yellow background in the manuscript.

We appreciate the reviewer’s recommendation to replace the optical microscope images in Fig. 3a. However, the more crystals were displayed and transformed in Fig.

3a, while only one crystal was shown in Fig. S29. We believe that Fig. 3a better demonstrates the universality of the crystal transformation. Therefore, we prefer not to replace Fig. 3a with Fig. S29 (0 h and 48 h). In addition, we have included Fig. S29 after Fig. 3a in the manuscript, with a reference to it for clearer representation of the transformation over different durations. We hope that these adjustments will provide readers with more clear and intuitive information.

As suggested, we have replaced Fig. S12 with Fig. S29 in the supporting information to more effectively illustrate the “in-situ” observation of the crystal-to-crystal transformation.

Comment 3.5: Solid-State Absorption Spectra (line 190):

The link to Fig. S18 is missing. Furthermore, the spectra presented are unusual, showing nearly continuous absorption without distinct maxima. The authors should explain this observation, especially since similar compounds exhibit a strong and distinct band around 250 nm in solution exhibits in solution a very intense and distinct band around 250nm (almost 6 times higher than in UVA range). In this context, the method used to record the spectra (e.g., film or diffuse reflectance) should be described more explicitly in proper section of the manuscript in place of only reporting the brand mark of the used spectrophotometer.

Reply: Thank you for your insightful comments.

We have added the missing link to Fig. S18 in the revised manuscript and highlighted it with a yellow background for clarity.

The solid-state absorption spectra were measured under standard testing conditions with power. It is important to note that absorption spectra for solids and liquids typically differ due to variations in their electronic structure, the propagation mode of light, and the microstructure of the crystals or materials. In contrast, solution-phase absorption spectra generally display distinct absorption peaks due to simpler intermolecular interactions. The continuous absorption behavior observed in cocrystals has been widely reported in the literature, such as in *J. Am. Chem. Soc.* **2015**, *137*, 11038–11046 and *Angew. Chem. Int. Ed.* **2018**, *57*, 3963–3967. In response to your suggestion, we have explicitly described the method used to record the spectra in the revised manuscript.

Comment 3.6: Stating on line 241, The authors hypothesize that NEt₃ solvation drives the “Cocrystal-to-Polycrystal Transformation”. To support this, they could attempt to grow AIC and ACTC crystals via slow evaporation of AOTC solutions in pure NEt₃ or

NEt₃/acetonitrile mixtures.

Reply: Thank you for your valuable suggestion.

In response to the reviewer's recommendation, we conducted experiments to grow AIC and ACTC crystals via slow evaporation of AOTC solutions in NEt₃/acetonitrile mixtures. As shown in Fig. R4, ACTC cocrystals formed on the walls of the bottle, while AIC crystals appeared at the bottom. The observation confirming the cyclization reaction of AOTC in NEt₃ and the formation of polycrystal via slow evaporation NEt₃/acetonitrile mixture solution.

Additionally, we attempted to place AOTC cocrystals in liquid pure triethylamine and slowly evaporate the solvent (Fig. R4e). This led to the formation of colorless, transparent microcrystals emitting blue fluorescence, alongside non-emissive TCNB or NEt₃·HCl crystals. This suggests that after the cyclization reaction, AIC molecules dissolved in triethylamine and recrystallized upon solvent evaporation. The solid-state fluorescence spectrum of the product shows blue luminescence at 439 nm without any red luminescence (Fig. R4f), confirming that only single-crystal AIC was generated, without the formation of the cocrystal ACTC. This is likely due to the lower solubility of TCNB in triethylamine compared to AIC, preventing the latter from cocrystallizing with TCNB in solution. In contrast, within microdroplets formed by vapor, the solubility of AIC may decrease due to interfacial and size effects, promoting its cocrystallization with TCNB.

Fig. R4 (a) Images of the bottle wall. (b) Images of the bottle bottom. (c) Optical microscope images of ACTC cocrystals on the wall under room light (left) and under 365 nm UV irradiation (right). (d) Optical microscope images of AIC crystals at the bottom under room light (left) and under 365 nm UV irradiation (right). (e) Optical microscope images (left) and fluorescence microscope images (right) of AOTC cocrystals placed in liquid triethylamine, followed by slow evaporation of the triethylamine solution. (f) Solid-state fluorescence spectra of the product obtained by the slow evaporation of the triethylamine solution.

Comment 3.7: The manuscript suggests that fluorescence changes from dual to single emission for AIC (lines 205-209) due to $\text{NEt}_3 \cdot \text{HCl}$, although $\text{NEt}_3 \cdot \text{HCl}$ does not appear to be present in the crystals. The basis for assigning to AIC a dual emission at 443 and 461 nm should be clarified, as these could instead arise from vibronic structuring or aggregate formation.

Additionally, the emissive wavelength used for lifetime measurements in Fig. S19c should be specified.

Reply: Thank you for your valuable comment.

We speculated that the dual fluorescence emission of AIC arises from vibronic structuring rather than aggregate formation. To further investigate, we measured the fluorescence spectrum of AIC in different solvents (*n*-hexane, DCM, and CH_3CN at a concentration of 1×10^{-5} M), as shown in Fig. R6. The dual emission gradually weakens as the polarity of the solvent increases, suggesting that the monomeric AIC molecule exhibits dual emission, and this dual emission is strongly influenced by the solvent environment. The extremely dilute testing conditions (1×10^{-5} M) in the fluorescence spectrum measurements also rule out the possibility of aggregate formation, supporting the conclusion that the dual emission is not due to aggregate emission. Previous reports have shown that the emission spectrum of anthracene derivatives is highly sensitive to environmental factors such as solvent polarity and pressure (*J. Mater. Chem. C* **2023**, *11*, 4892; *Chem. Eur. J.* **2010**, *16*, 3699).

In addition, as per the reviewer's suggestion, we have specified the emissive wavelength used for lifetime measurements in Fig. S22c and S22d in the revised manuscript. The revisions have been highlighted with a yellow background.

Fig. R6 Fluorescence emission spectra of AIC in *n*-hexane, DCM and CH₃CN (1×10^{-5} M).

Comment 3.8: I was surprised to read line 309 in material section that “all solvents were dried over Na”. Please Clarify which solvents were dried using such method. Indeed, they report the used of DCM as solvent to prepare AIC but the addition of reactive metals such sodium to chlorinated solvents can result in strong safety disappointment. Similarly, for acetonitrile (used for crystallization) is known to react with alkali metals.

Reply: Thank you for your valuable comment. We apologize for the confusion caused by our previous statement. To clarify, all the anhydrous solvents used in this study were purchased from commercial suppliers (Aldrich, Alfa, TCI, Daicel, etc.) and were used as received without any additional drying over sodium or other reagents. The mention of solvent drying over Na was an error in the manuscript, and we have corrected this in the revised version. The revisions have been highlighted with a yellow background for clarity.

Response Letter to Reviewers

Response to the Comments from Reviewer 3:

The manuscript entitled "Triethylamine Vapor-Induced Cyclization Reaction in Cocrystals Leading to Cocrystal-to-Polycrystal Transformation", submitted by Ben Zhong Tang and co-workers, is a revised version of a manuscript submitted a few months ago by the same authors, which I previously reviewed.

In this version, the authors have addressed almost all of my concerns described in my previous report, and as a consequence, I support its publication in Nature Communications. Nevertheless, some minor points (listed below) need to be corrected prior to publication, without necessarily requiring a further review process.

Concerning comment 3.1 regarding the triethylamine hydrochloride salt, I am pleased that the authors have followed my recommendation to include the full ^1H NMR spectra. However, the modification of the text with the insertion of the sentence:

^1H NMR spectrum of AOTC+NEt₃ also displayed a signal corresponding to equivalent Et₃N·HCl, with proton signals at 10.20, 1.20, and 3.05 ppm, revealing the formation of the salt mixed with the crystals (Fig. S13)."

could lead readers to believe that only one triethylamine molecule is sufficient to convert one AOTC unit (involving two BOX units and one TCNB). The ^1H NMR signal of an ethylenic proton of the BOX derivative (a doublet around 6 ppm, see ref. 50) is a very convenient way to determine the open/closed state ratio and the cis/trans conformation of numerous BOX derivatives. Based on its integration (0.96) and that of the triethylamine hydrochloride salt signal at 10.2 ppm (0.99), we can conclude a 1:1 ratio between the BOX unit and HNEt₃. Additionally, the 2:1 ratio between BOX and TCNB is also confirmed, as the integration of the singlet at 9.03 ppm (0.84) corresponds to the two aromatic protons of TCNB.

Reply: Thank you for your valuable comment. In response to the suggestion, we have revised the sentence for clarity as follows: " *^1H NMR spectrum of the AOTC+NEt₃ also indicated the simultaneous formation of Et₃N·HCl, as evidenced by proton signals at 10.20, 3.05 and 1.20 ppm (Fig. S13).*" The revised sentence is highlighted with a yellow background in the manuscript.

Concerning comment 3.3 regarding Information Encryption, I am pleased that the authors have performed the requested complementary experiments in solution. Despite their assumption about the formation of AOTC crystals on paper, I continue to support the inclusion of these data (in the supporting information, with a brief mention in the main text). This is motivated by

-First, no experimental data (PXRD, TEM images, etc.) demonstrate the effective crystallization of AOTC in a cellulosic medium.

-Second, the absence of such unusual chromic phenomenon in solution seems to support the crystallization hypothesis while ruling out the possibility of complex formation in solution.

Reply: Thank you for your valuable suggestion. In response, we have included the UV-visible titrations of AIO and AIC solutions with TCNB in acetonitrile in the Supporting Information for support the crystallization on paper (Fig. S35). Furthermore, we have revised the manuscript to briefly mention these solution-phase experiments in order to improve the quality and clarity. Specifically, we have added the following statement:

"Notably, UV-visible titration experiments revealed that the interaction between the two components in solution is relatively weak, with no analogous chromic phenomena observed in the solution state. This effectively rules out the possibility that the observed phenomenon arises from complex formation between TCNB and AIO/AIC under soluble conditions, especially considering that all subsequent operations in our study were conducted after drying the encrypted paper samples (Fig. S35). Therefore, the information encryption behavior can be attributed to the properties of the cocrystalline solids formed on the cellulosic paper."

All revisions have been highlighted in yellow in the revised manuscript.

Figure S35. a) UV-vis spectra of AIO in acetonitrile solution (0.1 mM) with varying amounts of TCNB (0–3.0 equiv.). b) UV-vis spectra of AIC in acetonitrile solution (0.1 mM) with varying amounts of TCNB (0–3.0 equiv.). c) UV-vis spectra of the AIO/TCNB complex formed by mixing different molar ratios of AIO and TCNB in acetonitrile solution (0.1 mM). d) UV-vis spectra of the AIC/TCNB complex formed by mixing different molar ratios of AIC and TCNB in acetonitrile solution (0.1 mM). The UV-visible spectra showed minimal changes, indicating weak CT interactions in solution (Fig. S35a and S35b). Due to the weak nature of these interactions, Job’s plot method could not reliably determine the binding ratio between AIO/AIC and TCNB in solution (Fig. S35c and S35d).

Concerning comment 3.5, the authors did not fully address my concern. While I completely agree with them that a strong variation could be observed between the UV-Visible spectra in solution and in the solid state. However, this continuum in the UVA region could also result from the experimental setup. In this context, my recommendation was to specify the method used to acquire the UV-Visible spectra of the different solid materials. Consequently, the insertion of the sentence: “UV-Vis

absorption spectra were recorded on a Shimadzu UV-Visible spectrophotometer and measured under standard testing conditions with solid” (line 373) is not sufficient. The model (such as example SolidSpec-3700i ??) and the type of performed measurement (was it transmittance or diffuse reflectance of the solid material? Or was it transmittance of the powder confined between two quartz/glass plates?) should be specified

Reply: Thank you for your professional suggestions. The UV-Visible spectrophotometer model is Shimadzu UV-2700 with an ISR-2600 Integrating Sphere Attachment, and the measurements were performed using diffuse reflectance of the solid materials. As suggested, we have specified the method used for acquiring the UV-Visible spectra of the different solid materials in the supporting information to enhance the accuracy.

Concerning comment 3.7, I continue to support the removal of the term dual emission from this manuscript, as it is not fully appropriate in this context.

As far as I am concerned, the term dual emission generally refers to the presence of two radiative de-excitation pathways, which are explained in terms of excited states of different nature (J. Phys. Chem. C 2013, 117, 35, 18154), such as in the case of luminescence from excimers, twisted intramolecular charge transfer (TICT) states, etc.

By contrast, describing the emission band as having a vibronic structure seems more accurate in this case, as it implies the presence of only one type of excited state. As clearly demonstrated by the authors, the observation of the vibronic structure in the AIC emission band—attributable to the anthracene moiety—is strongly influenced by the surrounding medium. Generally, anthracene excimer luminescence is observed at much longer wavelengths.

Reply: We thank the reviewer for raising the important point. We completely agree that the term ‘dual emission’ generally refers to the presence of two radiative de-excitation pathways. Therefore, we have removed the term ‘dual emission’ from both the manuscript and Supporting Information. The corresponding revisions have been highlighted with a yellow background in the manuscript for clarity.